


# Elemental composition of ambient aerosols measured with high temporal resolution using an online XRF spectrometer

Markus Furger[1,*], María Cruz Minguillón[2], Varun Yadav[3], Jay G. Slowik[1], Christoph Hüglin[4], Roman Fröhlich[1], Krag Petterson[3], Urs Baltensperger[1], André S. H. Prévôt[1]

[1]Laboratory of Atmospheric Chemistry, Paul Scherrer Institute, 5232 Villigen PSI, Switzerland
[2]Institute of Environmental Assessment and Water Research (IDAEA), Consejo Superior de Investigaciones Científicas (CSIC), Jordi Girona 18-26, 08034 Barcelona, Spain
[3]Cooper Environmental Services (CES), 9403 SW Nimbus Avenue, Beaverton, OR 97008, USA
[4]Laboratory for Air Pollution / Environmental Technology, Empa, Überlandstrasse 129, 8600 Dübendorf, Switzerland

*Correspondence to: Markus Furger (markus.furger@psi.ch)

**Abstract.** An Xact 625 ambient metals monitor was tested during a three-week field campaign at the rural, traffic-influenced site Härkingen in Switzerland during summer of 2015. The objective was to characterize the handling and operation of the instrument, evaluate the data quality by intercomparison with other independent measurements, and test its applicability for aerosol source quantification. The Xact was configured to measure 24 elements in $PM_{10}$ with 1-h time resolution. Hourly element concentrations ranged from a few ng m$^{-3}$ for trace elements in background conditions to tens of µg m$^{-3}$ for major elements during a high-emission event (fireworks). The total Xact element mass comprised approximately 20 % of the total $PM_{10}$ mass. The six major elements Si, S, Cl, K, Ca, and Fe contributed 95 % to the Xact $PM_{10}$ mass, the remaining 5 % were attributed to the trace elements. Data quality was evaluated by intercomparison with 24-h $PM_{10}$ filter data analysed with ICP-OES for major elements, ICP-MS for trace elements, and gold amalgamation atomic absorption spectrometry for Hg. 10 elements (S, K, Ca, Ti, Mn, Fe, Cu, Zn, Ba, Pb) showed an excellent correlation between the compared methods, with r$^2$ values ≥ 0.95, even though the Xact 625 yielded approximately 28% higher elemental concentrations than ICP for these elements. These elements demonstrate the high precision of the Xact instrument. An average 28 percent difference to ICP analyses might in part be attributed to the differences in the sampling systems (inlets), the geographic distance between the inlets and between the inlets and the freeway, and to uncertainties in the different analysis methods. 10 additional elements (Cr, V, Co, Ni, As, Se, Cd, Sn, Hg, Bi) could not be compared to a reference, because their concentrations were close to or below the minimum detection limits of at least one of the analysis methods. Sb revealed a calibration issue with the Xact, which requires correction. Si, Cl and Pt were not analysed with ICP, and thus could not be evaluated. The well-quantified elements were further used for a simple investigation of sources. The field campaign encompassed the Swiss National Day fireworks event, providing increased concentrations and unique chemical signatures compared to non-fireworks (or background) periods. Fireworks and traffic or rural background emissions could clearly be identified with their element mixture. The results demonstrate that multi-metal characterization at high-time resolution capability of Xact is a valuable and practical tool for ambient monitoring, exhibiting significant advantages compared to traditional elemental analysis methods.

## 1    Introduction

The quantification of trace elements in airborne particulate matter (PM) can be achieved with various techniques, such as inductively-coupled plasma mass spectrometry (ICP-MS), inductively-coupled plasma optical emission spectrometry (ICP-OES), X-ray fluorescence spectrometry (XRF), and proton-induced X-ray emission spectrometry (PIXE). These methods historically required a two-step procedure, i.e. sample collection followed by laboratory analysis. Ambient pollutants are typically collected on filter substrate for large time duration such as 8-h or 24-h sampling time to ensure that sufficient elemental mass is available for analytical analysis. In contrast to the non-destructive XRF method, sample preparation for





ICP analysis is very laborious, and the samples are destroyed during this process. XRF method has been successfully applied to aerosol characterization in the last decades. Measurement of low sample mass typically requires access to a synchrotron or similar X-ray facility (Bukowiecki et al., 2005; Bukowiecki et al., 2008; Calzolai et al., 2010; Calzolai et al., 2015; Lucarelli et al., 2011; Richard et al., 2010; Visser et al., 2015b; Yatkin et al., 2016), which is notoriously difficult due to the overwhelming demand for analysis time at such facilities. Technical advances in X-ray sources and detectors have recently resulted in the development of commercial systems capable of sampling and analysing ambient PM samples in sub-hourly or hourly resolution in quasi real time. Sampling with high time resolution generates large quantity of data capable of capturing source emission patterns occurring at shorter time duration. For source apportionment of PM components like elements, a high time resolution of the order of 1 hour or less is advantageous, as temporally variable environmental factors such as wind direction and speed or insolation may affect transport and source processes (e.g. resuspension, Sánchez-Rodas et al., 2007; Sarmiento et al., 2007; Visser et al., 2015b; Yadav and Turner, 2014). One such instrument, Cooper Environmental's Xact® 625 Ambient Metals Monitor, performs in-situ automated measurements of ambient $PM_{10}$ or $PM_{2.5}$ elemental concentrations for a user-defined set of 24 or more elements with a user-selected sampling time resolution of 15 to 240 minutes. The instrument is transportable and can be deployed in field campaigns where a suitable shelter with electric power and an appropriate sampling line connecting the outdoor with the indoor is available. Remote access to the data is possible during operation, which allows for a continuous monitoring of the operation status and the ambient metal content. An in-depth evaluation of the forerunner instrument Xact 620 was previously published by (Park et al., 2014).

An Xact 625 monitor was deployed for a month to test the handling and data production of the instrument. A small field campaign was organized at a monitoring station of the Swiss Air Pollution Monitoring Network (NABEL), where quality-controlled air pollution measurements are performed continuously. The NABEL network provided the reference for various data intercomparisons (Hueglin et al., 2005; Lanz et al., 2010). Comparisons between SR-XRF and filter samples analysed with ICP-OES and ICP-MS have been performed previously (Richard et al., 2010). The goals of this technical note are 1) to characterize the measurements of the test period in Härkingen and compare them with previous studies in Switzerland and elsewhere; 2) to examine the achieved data quality for the selected elements with respect to their minimum detection limits; 3) to quantify the measurement quality based on intercomparisons between the Xact and NABEL $PM_{10}$ data (1-h TEOM data and 24-h filter samples) for Härkingen; 4) to evaluate the applicability of the instrument at high time resolution in typical summer conditions and concentration ranges at a traffic-influenced rural site in Switzerland; and 5) to gauge the advantages of high time resolution sampling for a preliminary investigation of sources based on enhancement ratios and diurnal variability of elements. A pollution episode captured during the campaign resulted in high ambient concentrations, widening the range of studied concentrations. The selected elements represented a typical mix of elements at the selected site. In addition, a few elements notoriously difficult to measure in Switzerland due to their generally low ambient concentration were included, namely Ni, As, Pt, and Hg.

## 2    Experimental setup

### 2.1    Site characteristics

The field campaign was performed at the permanent station Härkingen (47.311877° N, 7.820453° E) of the Swiss Air Pollution Monitoring Network (NABEL, http://www.bafu.admin.ch/luft/00612/00625/index.html?lang=en ) from 23 July until 13 Aug 2015. This station is located next to the A1 freeway, the main traffic route between eastern and western Switzerland. About 1 km to the west the A2 freeway branches off towards the north. The local terrain is level and the traffic flows freely even during rush hours, limiting incidences of excessive braking or forced acceleration. There are villages to the south and east of the site, and agricultural land immediately to the west and north. Other local activities include industrial buildings approximately 500 m to the northwest (logistics businesses), and a metal processing company to the southeast





across the freeway. The site is well documented with respect to gas phase traffic emissions, PM number concentrations and particulate elemental carbon (EC, Hueglin et al., 2006), but an in-depth local source apportionment has not been realised so far except for organic aerosols measured in May 2005 (Lanz et al., 2010).

The first week of the campaign was characterized by lower temperatures (maximum temperature <25 °C, except 23 July)
with some occasional rain cleansing the atmosphere. The remaining two weeks were part of a summer heat wave, with temperatures reaching 36.4 °C at the maximum, and values above 30 °C on 7 days of this period. Only one precipitation event occurred during the hot period. The Swiss National Day (1 August) fell on a Saturday, and the weekend weather promoted outdoor barbeques and fireworks. The bulk of the fireworks were burnt on 1 August after 2200 CET, but some individual fireworks were also burnt on 31 July, and 2 and 3 August.

For this study, the Xact 625 monitor and a Q-ACSM (quadrupole aerosol chemical speciation monitor, Aerodyne Inc.) were installed in an air-conditioned trailer parked next to the NABEL station. This trailer was placed to the north of the freeway at ~23 m away from the centre of the freeway.  This placed the trailer on the orthogonal transect between the freeway and NABEL shelter, which is located ~27 m from the centre of the freeway. The trailer's instruments were connected to the NABEL station's power grid and Ethernet.

**2.2      NABEL instrumentation**

The NABEL station is equipped with a broad range of air quality instrumentation and standard meteorological sensors. The relevant instruments for this field test were the Digitel DA-80H HiVol sampler with a DPM 10/30/00 inlet for 24-h $PM_{10}$ samples collection, and a TEOM FDMS 8500 (Tapered Element Oscillating Microbalance, Filter Dynamics Measurement System, Thermo Scientific) for continuous (10-min) $PM_{10}$ mass concentration measurements. Standard meteorological
measurements such as temperature, wind speed and direction, and precipitation records are also monitored at this station. Furthermore, the station also provided hourly traffic counts for the freeway in the form of total number of vehicles, number of heavy duty vehicles (HDV), and number of light duty vehicles (LDV).

**2.3      Xact 625**

The Xact 625 ambient metals monitor (Cooper Environmental Services (CES), Beaverton, OR, USA) is a sampling and
analysing X-ray fluorescence spectrometer designed for online, semi-continuous measurements of elements in aerosols. In this study, ambient air was sampled with a flow rate of 16.7 actual lpm (i.e. temperature and pressure corrected) through a $PM_{10}$ flow separator (Tisch Environmental, TE-PM10-D) and the PM collected onto a Teflon filter tape. The flow is maintained to within about 1 %. After each sampling interval the filter tape is moved into the analysis area of the spectrometer, where it is illuminated with an X-ray tube in three excitation modes (25 kV and an Al filter, 48 kV and a Pd
filter, and 48 kV and a Cu filter), and the excited X-ray fluorescence is measured with an X-ray detector in the energy range from 0 to 40 keV. During this XRF analysis, the next sample is collected on a clean spot of the filter tape. This cycle is repeated during each sampling interval, which was configured to be 60 minutes for this study. After each analysis interval, raw and calibrated (for the actual volume in units of ng m$^{-3}$) concentration data was stored on the hard disk of the control unit. Daily advanced quality assurance checks (QA energy calibration test, QA upscale test) were performed during 30 min
after midnight to monitor shifts in the calibration. Thus, the sampling interval following midnight was limited to 30 min only.

The instrument was configured to quantify 24 elements (Si, S, Cl, K, Ca, Ti, V, Cr, Mn, Fe, Co, Ni, Cu, Zn, As, Se, Cd, Sn, Sb, Ba, Pt, Hg, Pb, Bi, plus Pd for QA). Each of these elements was calibrated individually with a reference sample. An uncertainty of 5 % or less due to fitting errors and uncertainties in the standards has been derived from laboratory
experiments with NIST standards (benchtop XRF, filter analyses). The uncertainty may be higher for concentrations close to the minimum detection limit (MDL), and due to self absorption effects for the lightest elements (Si, S, Cl, K, Ca). MDLs for





1-h sampling for each element are listed in Table 1. CES calculates MDLs using the sensitivity of the element and the counts in the region of interest of a blank unsampled section of tape, from where one sigma interference free detection limits are reported. XRF based MDLs are inversely proportional to the square root of the X-ray analysis time so that Xact MDLs are lower for longer sampling durations (Currie, 1977). The Xact reports purely elemental mass concentrations, and unless

otherwise noted (e.g. $SO_4^{2-}$) we refer to these pure elemental concentrations.

### 2.4     Q-ACSM

A Q-ACSM (Aerodyne Inc., Billerica, MA, USA) was operated in the trailer housing the Xact 625 during the campaign (Crenn et al., 2015; Ng et al., 2011). The Q-ACSM determines quantitative mass spectra of non-refractory particles up to mass to charge ratios (*m/z*) of 150. Ion fragments were attributed mainly to organic aerosols, nitrate, sulphate, ammonium,

and chloride, which comprise the reported data used in this study. The collection efficiency (CE) was determined for each spectrum according to Middlebrook et al. (2012), and its distribution peaked at the mode of 0.62 (±0.11). 293 out of 1055 CE values of the full ACSM dataset were equal to 0.45. The Q-ACSM collected sub-micron ($PM_1$) particles and chemically analysed them in 30-min intervals, which were aggregated to 1-h averages for comparison with the Xact 625 data. All concentrations used in this study were CE corrected.

### 2.5     24-h $PM_{10}$ filter samples

The 24-h $PM_{10}$ samples collected by the HiVol sampler on quartz filters were weighed at Empa laboratory in Dübendorf, Switzerland to determine the gravimetric daily $PM_{10}$ concentrations. These values were then used to correct the TEOM $PM_{10}$ concentrations on a daily basis. Therefore the 24-h TEOM values correspond to 24-h gravimetric $PM_{10}$. Ten 24-h $PM_{10}$ samples were analysed for their elemental composition at IDAEA-CSIC laboratory in Barcelona. A quarter of each filter was

acid digested using a mix of $HF:HNO_3$ (2.5:1.25 mL), the solution was kept in a Teflon reactor at 90°C for at least 6 h, and after cooling 2.5 mL of $HClO_4$ were added. The acid solution was brought to evaporation and the dry residual was re-dissolved with $HNO_3$ and diluted with milli-Q water for subsequent ICP-OES and ICP-MS analysis. This method has been validated and used in many studies, and is discussed in detail elsewhere (Minguillón et al., 2012; Querol et al., 2001; Querol et al., 2008). A total of 41 elements from Li to U were analysed: the major elements Na, Mg, Al, P, S, K, Ca, Ti, and Fe with

ICP-OES; the trace elements with ICP-MS. Si, Cl, and Pt were not analysed on the filters. Analyses of the reference material NIST 1633b (constituent elements in coal fly ash) using the same methodology as that for the samples yielded satisfactory results, with approximately 100% recoveries for the elements under study. Tests of the used methodology with respect to other ICP sample preparation and analysis methods, and applications of the methodology to NIST standards indicated the reliability of the method, exerting a maximum scatter of 10 % for any of the elements, with most uncertainty values clearly

below this limit. Relative uncertainties (precision) of the ICP measurements are less than 5 % for the elements with concentrations well above their respective detection limits, whereas the overall uncertainty reflecting the entire sampling procedure, the digestion and the ICP analysis is on the order of 25 %. Minimum detection limits for ICP were determined according to Escrig et al. (2009), and the values for the elements relevant for our intercomparison are listed in Table 1. Hg was analysed with a Hg gold amalgamation atomic absorption analyser (AMA-254, LECO instruments, Botasini et al., 2013;

Diez et al., 2007). The three methods are referred to as the offline or ICP methods (ICP-OES, ICP-MS), and the Hg gold amalgamation atomic absorption spectrometry is abbreviated with AuAAA in this paper.

Three of the 10 filters were also analysed with a benchtop XRF system by CES, and with ICP-MS by an independent lab (Eastern Research Group, ERG, Research Triangle Park, NC, USA) to investigate inter-laboratory scatter. ERG used a different digestion method than IDAEA-CSIC. In addition, three filters were prepared with a reference aerosol of known

concentration for Fe, Cu, Zn, Sr and Pb, which then were analysed by CES, IDAEA-CSIC, and ERG, again to gain insight into the inter-laboratory scatter. Details on these data and the methods are given in the supplement of this article.





## 2.6    Data coverage and synchronization

The Xact 625 measurements started on 23 July 2015 1200 CET, and ended on 13 August 2015 0600 CET. The sampling interval was set to 1 h. Two interruptions occurred during the sampling period: one due to an Xact 625 computer problem (33 h), the other one due to a delayed filter tape change (10 h). The Xact dataset consists of 456 valid 1-hour samples out of 499 possible samples, attaining a coverage of 91.4 %. The NABEL data were tailored to coincide with the Xact data. The 10-min TEOM $PM_{10}$ values were aggregated to 1-h values to synchronize them with hourly Xact 625 measurements. Additionally, TEOM data were also adjusted to the gravimetrically determined $PM_{10}$ masses from the HiVol filters to provide an independent reference for intercomparisons. The data used here contained some gaps which were only partly synchronous for the selected parameters. Wind speed and direction missed 12 data points (2.4 %), precipitation 26 (5.21 %), and $PM_{10}$ 53 (10.6 %) at hourly time resolution. The ACSM data contained a gap of 14.5 h due to an erroneous DAQ value, which caused the data to be very noisy for that short period. These values were rejected from the analyses, and only the remaining 972 data points were averaged to 465 1-h values, which then were resampled to the 456 Xact data points.

For the comparisons of the different instruments and sampling intervals, all data were resampled to the corresponding times of the Xact 625, according to the sub-classifications of the data set (e.g. according to wind sectors). For the intercomparisons with the 10 filter samples, the Xact data of the corresponding days were averaged to the 24 h of the filter samples. During each 24-h period, Xact generated 23 1-h values and 1 30-min value were aggregated to 24-h daily averages. This procedure implicitly assumes that the half-hour sample of the first sampling hour is representative for the whole hour. Tests with a 23.5-h weighted average yielded differences of less than 3 % between the two calculation methods.

## 3    Results and discussion

### 3.1    Data validity derived from general statistics and minimum detection limits

The complete Xact dataset is visualized in Figure 1, and general statistics are given in Table 1. The salient feature of the concentration time series is the huge peak late on 1 Aug, caused by the National Day's fireworks episode. Further peaks before and after that day warranted dividing the full data set into a fireworks period and a non-fireworks period. The fireworks period started on 31 July 2015 2200 CET and lasted until 4 August 2015 1100 CET, as will be discussed in more detail in Sect. 3.3. The remaining non-fireworks period is representative for the typical background concentrations at Härkingen, and can be compared to literature values.

MDLs for the Xact 625 and for ICP-OES/MS and the Hg AuAAA method are listed in Table 1. Note that MDLs of elements measured by Xact 625 are based on 1-hr sampling time while MDLs of filter based elemental concentrations are based on 24-hr. Generally, values below 3*MDL are expected to have much higher uncertainty. Hence, elements with more than 80 % of the data below 3*MDL were rejected from further examination. Xact 625 MDLs have not been determined by the manufacturer for Si, S, and Cl, because self-absorption effects for elements lighter than Ca become more important with decreasing atomic number (Formenti et al., 2010). However, these three elements are abundant, and we assume that they are well above their Xact detection limit. For these elements, an ICP MDL is only given for S, because Si cannot be determined in the filter samples, as it is a main constituent of the quartz filters and is also digested during sample preparation, and Cl cannot be determined by ICP. The table indicates the amount of data points >MDL in percent for the different analysis methods. The elements K, Ca, Ti, Mn, Fe, Cu, Zn, Sn, Sb, Ba, and Pb have most values above the MDL, and their measurement should thus be reliable. Seven Xact elements have >50 % of their data points below MDL and more than 90 % below 3*MDL: V, Co, As, Se, Cd, Pt, and Bi. Cr and Cd show the same behaviour for ICP. Ni revealed variable blank concentrations in the filters and could therefore not be reliably measured with ICP. Hg is also mostly below MDL in the AuAAA measurement, and Pt was not measured with ICP at all. In summary, 12 elements are above their MDLs for both the





XRF and the offline methods, 7 elements are below MDL for the XRF, and 4 elements are below MDL for the offline methods (of which only Cd is below MDL for both XRF and ICP).

The comparisons between online Xact 625 and offline 24-h $PM_{10}$ elemental concentrations for 21 elements are shown in Fig. 2, Fig. 3, and Table 2. Only the 21 elements analysed by both methods are compared by dividing them into two groups based on data characteristics.

Group A shows excellent correlations between the two measurement methods ($r^2$ values >0.95) and only small intercepts (<40% of mean concentration), and consists of the elements *S, K, Ca*, Ti, *Fe*, Mn, Cu, Zn, Ba, and Pb (elements in *italics* were analysed with ICP-OES; the others by ICP-MS). On average, the Xact 625 yielded approximately 28% higher elemental concentrations than ICP for the Group A elements.

The high linearity and little scatter in the regressions testify for the precision of both the Xact and the ICP methods, but the differences in the slopes for different elements require further investigation. No systematic deviations based on elemental molecular weight or X-ray energy conditions were observed in these slopes. The deviations of the slopes from unity may be partially attributed to the different inlets for the Xact and the HiVol samplers (Panteliadis et al., 2012), which may produce a difference in collected mass on the order of 10 %. The inlets were not explicitly tested for their cut-off characteristics in this study. A slightly different cut-off value for the particle size may lead to differences in the collected mass, especially for the largest and heaviest particles in $PM_{10}$, and hence to an underestimation or overestimation of the total mass collected with a particular inlet. This may be of special relevance in a near-road setting with lots of re-suspended dust (ACES, 2012).

The results of additional investigations of a few selected filters by independent labs and analytical methods for understanding these differences are discussed in the supplementary material. Examination of reference samples indicated a high precision in XRF measurements despite consistently underestimating the absolute concentration (range of underestimation varying between 6 to 14% depending on the element). In contrast, ICP measurements indicated greater variability (underestimation by 30% to overestimation by 60%, depending on the element) and hence higher uncertainty in estimated ambient concentrations. Examination of three filter samples collected during the campaign by an offline XRF instrument (by CES) and by ICP at an external laboratory (ERG) indicated a variability of about 30% for most elements.

The relative mean difference of 28 % between Xact and filter data (analysed with ICP) for samples collected during the field campaign appears to be systematic. Such differences may result from a difference in location of the Xact and filter sampling inlets (~5 m) and their relative distance from the freeway. Ultrafine particle number concentrations from dust resuspension due to vehicle traffic are known to decrease with increasing distance from the road, with the sharpest decline observed within the first 50 m (Hagler et al., 2009). Crustal elements, which dominate in the $PM_{10}$ size fraction, are expected to settle faster due to larger aerosol size. Hence the difference in Xact and ICP reported $PM_{10}$ elemental concentrations may be indicative of a gradient in PM occurring for some elements in close proximity to roadways. To quantify the different effects, a field campaign with a different design would be needed, e.g. an array of samplers along a line perpendicular to the freeway. However, since the difference is also observed for S, which is typically found in the fine mode, does not have a major traffic related source, and is not expected to suffer from incomplete digestion we assign part of the differences also to calibration issues in the Xact.

Group B, the remainder, consists of the elements V, Cr, Co, As, Se, Cd, and Bi, i.e. of those elements that are close to or below their Xact MDL, plus Ni, Cd, Sn, Sb and Hg. Ni, Cd and Hg were below MDL for the offline methods (Cd for Xact and offline methods). Although an intercomparison of these elements may not be justified, we observed some features in the regressions of the Group B elements in Fig. 2 that are worth commenting. Cr is below the ICP-MS MDL for 60 % and below 3*MDL for all filter values, but 75% are above the Xact MDL. The ICP measurement is below MDL because of the high and variable blank concentrations, which make a meaningful blank subtraction difficult, and which increases the Cr MDL in these samples. Although the slope of Cr is 1.23 and thus comparable to the other Group A slopes, a comparison with ICP values is statistically not robust. However, Cr seems to be quantifiable with the Xact. The regression plot of Bi shows



two extreme values on 31 Jul and 1 Aug corresponding to the fireworks days. These two points are above MDL for both methods and suggest good quantitative agreement between XRF and ICP for these two high-concentration cases. Sb shows a moderate correlation ($r^2 = 0.47$), and a large intercept. Sn behaves similarly as Sb, with an $r^2$ value of 0.15. The large intercepts hint toward a problem in processing the Xact blanks. In addition, when Ca is abundant, as in our case in

Härkingen, the Sb Lα line interferes with the Ca Kα line, producing low signal-to-noise ratios for Sb, and similarly for the pair K – Sn. Hence, the reported Xact concentrations of these two elements reflect mainly spectral noise. 60 % of the filter Hg data were below MDL and thus cannot be well compared with the Xact Hg data. Inspection of the Xact raw Hg spectra showed a possible interference from Br causing the fitting routine to attribute some Br mass to non-existent Hg peaks in the spectra. Br was not calibrated in the fitting routine. Thus the Hg concentrations reported by the Xact seem to be due to this

interference and are not realistic, even though 86 % of the measured values are above the Xact MDL. Values < 1.5 ng m$^{-3}$ are in the same order of magnitude as the fortnight values of Chiaradia and Cupelin (2000) for the city of Geneva (Table 3). To summarize, the Group B elements show issues with the minimum detection limits of at least one of the analysis methods, which made a comparison meaningless. Individual data points above MDL reveal nevertheless a usable quantification by the Xact in these particular cases. Sb, Sn and Hg showed instrumental problems (line interferences) for the Xact technique that

need to be improved.

The Group C elements Si, Cl, and Pt were not measured on the filters. An Xact MDL for S has not yet been determined. We can obtain information on the accuracy of the S measurement by comparing its concentrations with the concentrations of another element originating from the same source, or belonging to the same chemical compound. Hence, S (with unknown MDL) and K (with known MDL) concentrations were used to this end. K and S were highly correlated ($r^2 = 0.99$) during the

fireworks period, with a slope of 2.30 ± 0.05 for K vs. S concentrations, which agrees with the stoichiometric relation between K and S when forming $K_2SO_4$. For the non-fireworks periods the correlation was weak ($r^2 = 0.16$), which hints towards a completely different, more random relationship between the two elements, as expected. Based on these results, it can be concluded that S can be measured as reliably as K by the Xact, despite the lack of an established MDL for S. Xact Pt measurements were always below MDL, and no conclusion about the Pt accuracy can be drawn.

**3.2    Comparisons with other data**

Figure 1 shows that roughly 95% of the total analysed elemental mass by Xact is comprised of 6 elements: Si, S, Cl, K, Ca, and Fe. These major elements all show average concentrations >100 ng m$^{-3}$. Si, S, Ca, and Fe are observed throughout the study, although with high variability. Cl and K are abundant only episodically: Cl is strongly tied to westerly winds during the last week of July, and is practically absent after 2 Aug. K is prominent during the fireworks period. Ti, Cu, and Zn show

daily average concentrations between 11 and 34 ng m$^{-3}$. The other analysed elements were found in daily average concentrations <10 ng m$^{-3}$. The concentrations are of the same order of magnitude as those recently measured at other places in Switzerland, e.g. at an urban background site in Zurich (Minguillón et al., 2012; Richard et al., 2011, Table 3), but are generally lower than the measurements in older studies (Chiaradia and Cupelin, 2000; Gälli Purghart et al., 1990; Röösli et al., 2001, Table 3). The decreasing trends in PM and trace element concentrations have been documented in numerous

NABEL reports on the air quality in Switzerland (e.g., BAFU and Empa, 2015; Gianini et al., 2012). These trends make it preferable to use modern studies for comparisons. Furthermore, the episodic nature of the 2015 campaign also demands for some generosity when comparing the measured values with annual or seasonal mean values.

A time series of the Xact 625 total element concentrations together with the NABEL TEOM PM$_{10}$ data and the total ACSM non-refractory (NR)-PM$_1$ concentrations with 1-h resolution is presented in Figure 4. The total ACSM NR-PM$_1$

concentration is the sum of sulphates, nitrates, ammonia, chlorides and organic aerosols. Total PM$_{10}$ shows a generally increasing trend over the whole campaign, with a strong peak superposed on 1 Aug 2015, which coincides with the peak in the Xact data. The peak is due to the fireworks burnt on that evening. On average the Xact 625 elements make up about 20 %





of the total $PM_{10}$ mass (regression slope 0.2, $r^2 = 0.63$). A complete mass closure cannot be achieved, because the NABEL station only reports the total gravimetric $PM_{10}$ mass and $PM_{2.5}$ elemental carbon (EC) concentrations with diurnal or better time resolution.

The measurement accuracy for S was tested by comparison with ACSM sulphate measurements (Figure 5). The S

concentrations of the Xact 625 were multiplied with a factor of 3, assuming that all S occurred in the form of $SO_4^{2-}$. The slope of the regression line for the non-fireworks case is 1.34, with $r^2 = 0.85$, in line with the Group A elements, and in agreement with the comparison of S from Xact and from ICP, hence the slope between ACSM $SO_4$ and ICP $SO_4$ would be ~1. The high linear correlation suggests a high precision of the Xact 625 data, but does not allow a definitive answer on the accuracy because of expected self-absorption effects. The comparison for the fireworks period looks different. The scatter is

large, and the correlation coefficient is only 0.1. We hypothesize that fireworks produce larger and non-refractory particles (e.g. $K_2SO_4$) not measured by the ACSM.

In summary, the on average 25 to 30 % difference between the Xact and ICP data can probably be explained by differences in the sampling inlets, the distance between the inlets, and uncertainties of the different analysis methods. The correlation coefficients close to 1 for many elements demonstrate the high precision of the Xact and ICP methods. The obtained time

series of those elements can thus reliably be used for source apportionment. The subsequent analyses (e.g. elemental concentration ratios, enhancement ratios) were done with the unmodified Xact data. The only exception is estimation of a mass budget in the discussion of the extreme concentrations in section 3.3.

### 3.3    Extreme concentrations: the fireworks period

As mentioned in Sect. 3.1 the measurement campaign can be divided into a fireworks and a non-fireworks period. A K

concentration > 220 ng m$^{-3}$ served as the criterion to distinguish between these periods, and we required the fireworks period to be contiguous from the first increase in K on 31 July 2015 2200 CET to the final decay to background values on 4 August 2015 1100 CET. The average K concentration during the fireworks period was 2 µg m$^{-3}$, but this period showed extremely high hourly $PM_{10}$ concentrations and an element mix different from the remainder of our test campaign.

Figure 4 shows an extreme peak on 1 Aug 2015 2300 CET, when the NABEL $PM_{10}$ reached a 1-h concentration of 59.6 µg

m$^{-3}$. The Xact 625 monitor reported a total of 48.4 µg m$^{-3}$ for the sum of all analysed elements (except Pd, which was used only as an internal standard). The bulk of this concentration (47.4 µg m$^{-3}$) was made up of a few elements (in brackets: concentrations in µg m$^{-3}$, and abundances relative to total analysed element mass, $PM_{10\text{-element}}$): K (27.3, 56.5%), S (12.0, 24.9 %), Cl (4.5, 9.2 %), Fe (1.5, 3.2 %), Ba (1.1, 2.3 %), Si (1.0, 2.2 %). Absolute K and S concentrations are in good agreement with the values in Drewnick et al. (2006). K likely originated from $KNO_3$, a basic constituent of black powder (Drewnick et

al., 2006; Kong et al., 2015). The period was characterized by four strong peaks (with decaying intensity) in the K/S ratio (Fig. 6). The expected K/S ratio for black powder is 2.76 (Drewnick et al., 2006), and was nearly attained on the first two evenings.

The ACSM $NH_4$ did not show an effect of the fireworks in the time series (Figure 6); $NO_3$ and organic aerosol showed a quick drop immediately before due to a wind shift, and only $SO_4$ and chloride showed a five-fold and 2.6-fold increase,

respectively, at the time of maximum fireworks activity, relative to the pre-fireworks period, and a subsequent decay. The absence of fireworks $NO_3$ has been observed previously (Drewnick et al., 2006) and indicates that all nitrogen of the black powder is converted into $N_2$ or $NO_x$. The $NO_x$ option is not supported by a strong increase in $NO_x$ in the NABEL data, nor is there a strong correlation between $NO_x$ and K, and hence our measurements indicate the $N_2$ pathway. $SO_4$ reached a peak concentration of 5.9 µg m$^{-3}$, and chloride reached 0.5 µg m$^{-3}$. The organic aerosol concentrations showed a value of 3.2 µg

m$^{-3}$, and were further increasing after the fireworks. The other species showed minima during the hours before the fireworks, which coincides with northwesterly flow, followed by a slow increase and south-westerly flow over the next few hours. In total, the ACSM $PM_1$ contributed 11.9 µg m$^{-3}$ (25 %) to the aerosol concentration in the fireworks hour (2300 CET). The





$SO_4$ peaks coincide with a strong drop of the $PM_1$ $NO_3$ concentration, which indicates that $NH_4NO_3$ reacted with $H_2SO_4$, and $HNO_3$ escaped into the gas phase. The slight increase in ACSM chloride may be due to reaction of $NH_3$ with HCl.

The comparison of the Xact concentration and the TEOM $PM_{10}$ concentration for the 1 August peak in Fig. 4 requires taking into account the systematic difference between the Xact and TEOM measurements discussed above, and the fact that the

5 elements are typically not present in elemental but rather in their oxidized form, such that the mass of the latter needs to be included for a quantitative comparison. We therefore estimate a mass budget for the fireworks peak hour at 2300 h CET, when the six elements K, S, Cl, Fe, Ba, and Si comprise the bulk of the total mass. We calculate the ion balance for the positive ions $K^+$, $Fe^{2+}$, $Ba^{2+}$, $Si^{2+}$, submicron $NH_4^+$ and negative ions $Cl^-$, $SO_4^{2-}$, and submicron $NO_3^-$. We further add all available components from the NABEL station ($PM_{2.5}$ EC) and the ACSM ($NO_3$, $NH_4$, organic aerosols, but not $SO_4$, as this

is already considered with the S in the Xact data) to the mass balance. Using the measured values of the Xact 625 yields an excess of negative ions of 2.7 %, and a total mass of 77.6 µg m$^{-3}$, which overestimates the TEOM value of 59.6 µg m$^{-3}$ by 30 %. If we scale the Xact concentrations towards the ICP concentrations with the corresponding regression slopes from Table 2, using for Cl and S the average slope of 1.22 from the other four elements, then the calculation yields an excess of 11% in positive ions which are then assumed to be neutralized by oxygen. This yields a total mass of 63.7 µg m$^{-3}$, which is only 7 %

higher than the TEOM value. Our values are lower limits of the total mass, because the balance is incomplete with respect to relevant elements in the fireworks (e.g. Sr) and other chemical species like carbonaceous and nitrogen containing molecules in the coarse fraction. The result shows that the bulk of fireworks $PM_{10}$ aerosols are a few metals oxidised to sulphates, chlorides, and oxides. The result further demonstrates the applicability of the Xact in conditions with high concentrations, and the advantages of high time resolution sampling.

**3.4 Investigation of sources**

Trace elements can be excellent tracers for specific aerosol sources (e.g. Hopke, 2016; Park et al., 2014; Querol et al., 2007; Visser et al., 2015a). A simple approach for characterizing a common source for a group of elements is to study the temporal covariation of the elements in this group. For our Härkingen data, the time series indicate the strong influence of the fireworks on the concentrations of K, S, Ti, Cu, and Ba (Figures 7 and 8), which are important constituents of fireworks

(Kong et al., 2015; Moreno et al., 2007). In addition, we would expect Sr as a fireworks tracer (Kong et al., 2015; Moreno et al., 2007). We checked a few raw spectra from the fireworks and non-fireworks periods and could clearly identify an enhancement of Sr during fireworks, while the peak was definitively absent during the non-fireworks period. Sr was, however, not quantified in our configuration, as we put emphasis on crustal elements and some special trace elements difficult to detect in Switzerland (Hg, Pt). The gradual decay of the K/S ratio to ~0.5 over the fireworks period (Figure 6)

hints towards a depletion of K relative to S, which may indicate the increasing presence of secondary sulphate from $SO_2$ oxidation, or to the influence of a source other than fireworks. Barbecues are a typical summer evening activity, especially on weekends. The charcoal K/S ratio ranges from 0.3 to 2 (Humphreys and Ironside, 1980), which brackets our measurements.

The 1-hour sampling interval allows for the resolution of diurnal variations of the elements. Ca and Ba are presented in

Figure 7 and the other elements in the supplement (Fig. S3), which shows the classification of the data according to fireworks and non-fireworks periods. It can be seen for the time from 2300 h to 0600 h that the elements Ba, Bi, Cu, K, S, and Ti show a clear distinction between the two periods. The fireworks elements show a maximum concentration at 2300 h and a gradual decay over the next 6 to 12 h into the morning hours of the (following) day. Mn, Fe and Zn also show an increased and then decaying concentration after midnight, but the difference compared to non-fireworks days is within the

data variability. Si is depleted during the fireworks period relative to the non-fireworks period. This is probably a weekday vs. weekend effect, when fewer heavy duty vehicles (HDV) circulate (Switzerland does not permit HDV use on Sundays), and less road dust is re-suspended (Bukowiecki et al., 2009). For the non-fireworks cases the transition elements Mn, Fe, Zn,




and the element Pb are characterized by a broad morning peak with a maximum around 1000 h, correlating well with the increasing traffic in the morning hours, and the breakup of the temperature inversion. The non-fireworks Si increase in the morning hours runs parallel to the increasing traffic and $NO_x$ (Fig. 9). However, Si and traffic deviate in the afternoon, when traffic still increases until the evening rush hour, while both Si and total PM concentrations decrease. The Si curve thus

resembles more the number of HDV, which remains constant throughout the working hours, than the total number of vehicles (Bukowiecki et al., 2010).

To identify the fireworks tracers, an enhancement ratio (ER) was defined as the ratio between the mean concentrations of an element for the fireworks period to the concentrations in the non-fireworks period (Figure 8). For K, Cu, and Ba the ER is larger than 2 (Cu), and goes up to 10.6 (Ba). S, Cl, Ti, Zn and Pb show an intermediate ER between 1 and 2. Cr, Mn, and Fe

ER are close to 1. Si and Ca are depleted with an ER around 0.5, both of which are probably related to the above weekend effect. The elements with the high ER are clearly identified as elements of fireworks: S, K, Ti, Cu, Zn, Ba, Bi.

Further refinement of sources can be obtained when classifying the non-fireworks data by wind direction into a north (270° – 0° - 90°) and a south (90° – 180° - 270°) sector (Figure 10), with the south sector more strongly influenced by highway emissions (Hueglin et al., 2006). The freeway runs from 120 ° to 270 °, but a shift of the applied wind sectors by 20 °

showed no significant difference. The north sector characterizes the (rural) background concentrations of the central Swiss plateau. Table 1 summarizes the mean element concentrations for the campaign divided into the different periods and wind sectors. Ba, Cr, Cu, Fe, and Mn show the signature of continuous freeway traffic emissions during the day. Pb and Zn show a morning peak only and are well correlated in both sectors. Si, K, Ca, and Ti show another pattern that could reflect the local and regional transport of crustal material partly re-suspended by traffic (south sector), partly originating from the agricultural

area north of the freeway. S shows a high variability and no clear difference between the sectors, and an interpretation is difficult. This could be the result of a specific, perhaps episodic wind pattern advecting higher concentrations from south during the night and from north during the day. Cl also shows a unique behaviour. An increase in Cl was seen in the ACSM data only during the fireworks. We therefore conclude that Cl during the non-fireworks period was of regional (probably maritime) origin and hence rather an indicator of long-range transport. A full understanding of the Cl behaviour requires a

more detailed study of the wind field and a more sophisticated source apportionment which is beyond the scope of this study. Figure 11 shows the enhancement ratios south/north. Apart from Cl, all south – north differences are positive, and Si, S, Ca, and Fe concentration differences are larger than 80 ng m$^{-3}$. These are mainly crustal elements (although Fe is also emitted from vehicles). The enhancement ratios of the transition elements Cr, Mn, Fe, Cu, Zn, Ba, and Pb are larger than 1.2 and related to traffic emissions (engine abrasion, tyre wear, brake wear).

## 4 Conclusions

A three-week test of a Cooper Environmental Xact 625 Ambient Metals Monitor was performed at the Swiss NABEL station Härkingen. The instrument was configured to measure 24 elements continuously with 1-h time resolution. The selection of elements ranged from Si to Bi, thus covering a range of environmentally relevant elements. Besides the 'standard' elements from K to Pb, which have been well characterized by the manufacturer with respect to their accuracies and detection limits,

we included several abundant light elements (Si, S, Cl) and – more for curiosity - some low-concentration elements (As, Pt, Hg) in our selection to test the behaviour of the instrument in a typical Swiss environment. We tested the measurement quality of the Xact 625 by intercomparison with well-established methodologies (ICP-OES and ICP-MS analyses on 24-h $PM_{10}$ samples for major and trace elements, and AuAAA for Hg), ACSM, and TEOM, and used additional meteorological data for the interpretation of the results.

The general findings are:

- The total of elements analysed with the Xact comprised of approximately 20 % of the $PM_{10}$ mass.



- The Xact 625 produced element concentration time series that were highly correlated with the ICP analyses of 24-h filter samples ($r^2 \geq 0.95$), even though the slopes deviated from 1.

- Element concentrations ranged from ng m$^{-3}$ (in background conditions) to tens of µg m$^{-3}$ (during the fireworks), and no instability in operation due to sample overload or else could be observed.

- Measured concentrations agreed reasonably well with other recent field measurements in Switzerland.

The results for measurement accuracy, precision and data quality are:

- We found an excellent correlation between Xact 625 and ICP-OES/ICP-MS for the elements S, K, Ca, Ti, Mn, Fe, Cu, Zn, Ba, and Pb ("Group A"), indicating that all methods reproduce these concentrations well. Systematic differences of on average 25 to 30 % are attributed to physical reasons in the experiment settings, such as the

different characteristics of the two inlet systems, the distance between the inlets and to the main source (freeway), and uncertainties in the various analysis methods. For XRF this includes particle size dependent self-absorption effects for the lighter elements and line interferences between different elements. For ICP this includes the entire sampling, digestion and the analysis procedure, as indicated by limited inter-laboratory and inter-method comparisons).

- The remaining elements ("Group B") of the filter intercomparison, V, Cr, Co, Ni, As, Se, Cd, Sn, Sb, Hg, and Bi (11 elements) were mostly below detection limit of at least one method, or showed issues with the analysis procedures (Sn, Sb, Hg). A general quantitative statement on their quality could not be achieved. Notice here that a longer sampling time, e.g. 2 or 4 hours, would have lowered the Xact MDLs, but on the cost of a reduced time resolution.

- Si and Cl were not analysed on the filters, and their Xact detection limits have not yet been determined. Hence their accuracies could not be quantified directly. Some indirect approaches were calculated.

- The Pt concentrations reported by the Xact 625 were below MDL, and Pt was not analysed on the filters. No conclusion about the accuracy of this difficult to measure element can be drawn.

The results from the investigation of sources indicate:

- The period influenced by fireworks was clearly distinguished from the normal conditions. It showed extremely high or strongly enhanced concentrations of elements S, K, Cl, Cu, Zn, Ba, and Bi.

- The normal, non-fireworks conditions could be split according to wind direction into a traffic-influenced sector and a rural background sector. The enhancement of traffic-related elements relative to the background mix could clearly be shown.

- Average diurnal variations of element concentrations could be calculated. They further demonstrated the capabilities of the Xact 625 instrument to refine the investigation of PM sources.

Compared to rotating drum impactor sampling with synchrotron radiation induced XRF or streaker sampling with PIXE analysis, the Xact 625 measures ambient concentrations of the most relevant elements in near real time, and provides data with a delay of only one sampling/analysis cycle. This is a major advantage compared to the usual time delay of a couple

35  months caused by the restricted access to synchrotron or accelerator facilities. Of course, the high time resolution of the Xact 625 comes at the cost of sensitivity, visible in the minimum detection limits, which are higher than the MDLs for the offline methods. In our short-time test study, we fixed the sampling interval to 1 h, but longer sampling intervals and therefore lower MDLs could be set at the instrument, thus extending the number of successfully analysed elements. Another advantage is the continuous operation capability that circumvents sample number limitations due to restricted beamtime assignments at

synchrotrons. This enables long-term sampling and routine monitoring. Useful extensions of the present capabilities of the Xact could be the addition of more elements to be analysed (especially under the circumstance that the full mix of observed elements cannot always be known in advance), improved quantification of the lightest elements (especially their MDLs), a



vacuum or helium device for analysing light elements like Na and Mg, and an inlet switch to alternately measure $PM_{10}$ and $PM_{2.5}$ with one single instrument.

## 5    Supplementary Material

## 6    Competing interests

Krag Petterson and Varun Yadav are employed by Cooper Environmental Services, the manufacturer of the Xact® 625.

## 7    Acknowledgements

This study has been partly funded by the Swiss Federal Office for the Environment (FOEN). We thank René Richter and Roland Scheidegger of PSI for their support during the field campaign. We are grateful to Chris Koch and John Cooper of Cooper Environmental Services for instructions on instrument operation and numerous discussions on the results. Andrés

Alastuey, Xavier Querol and laboratory personnel from IDAEA-CSIC are also acknowledged. We also thank Julie Swift and Randy Mercurio of ERG for the ICP-MS analyses.

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





## 9 Figures

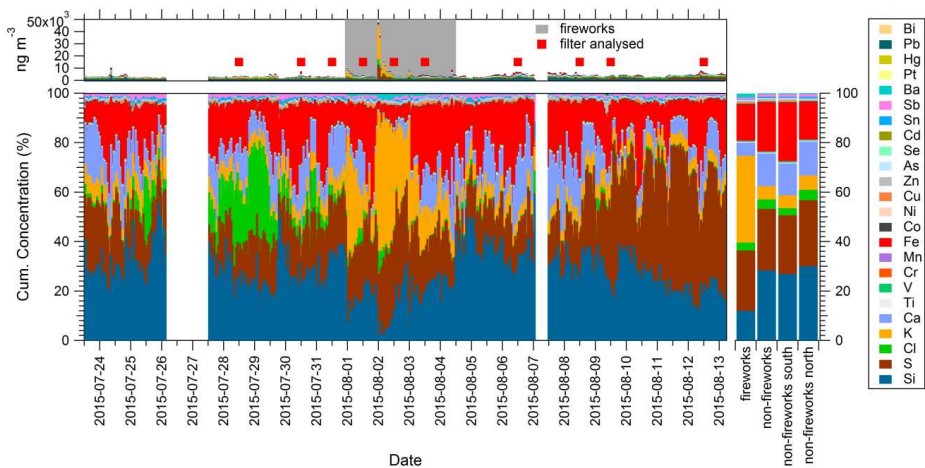

5 **Figure 1: Main panel: Relative amount of analysed elements by Xact 625 during the field campaign. Top panel: Absolute concentrations, stacked. The grey shaded area denotes the fireworks period. The red squares mark the days when 24-h filters were analysed and used for comparisons in this study. Bottom panel: relative cumulative elemental concentrations, stacked. Right panel: Average relative contributions (in %) of elements for the fireworks period, the non-fireworks period, and for the south and north sectors during the non-fireworks period.**





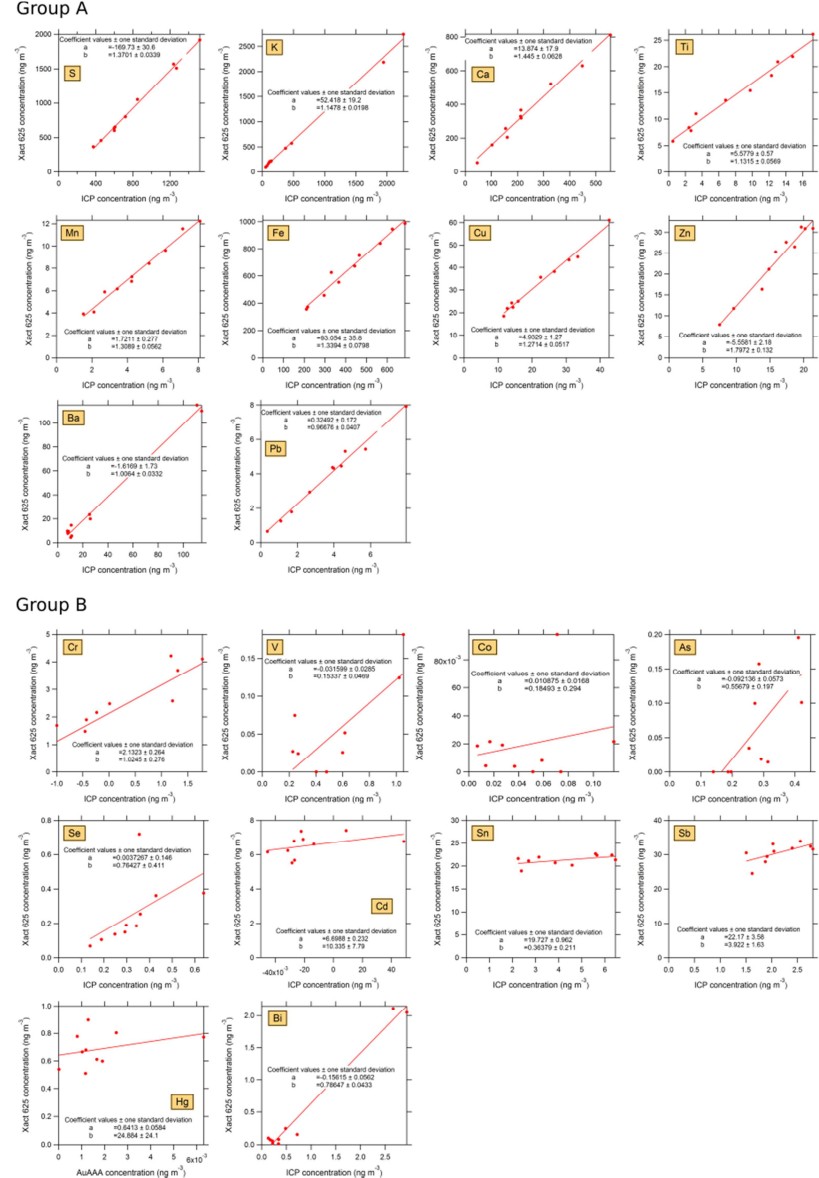

**Figure 2: Scatterplots of Xact 625 (ordinate) vs. ICP-OES/MS (abscissa) data. The Levenberg-Marquardt linear least squares fitting method was applied, taking the ICP measurements as the independent data. Regression equation is y = a + bx.**



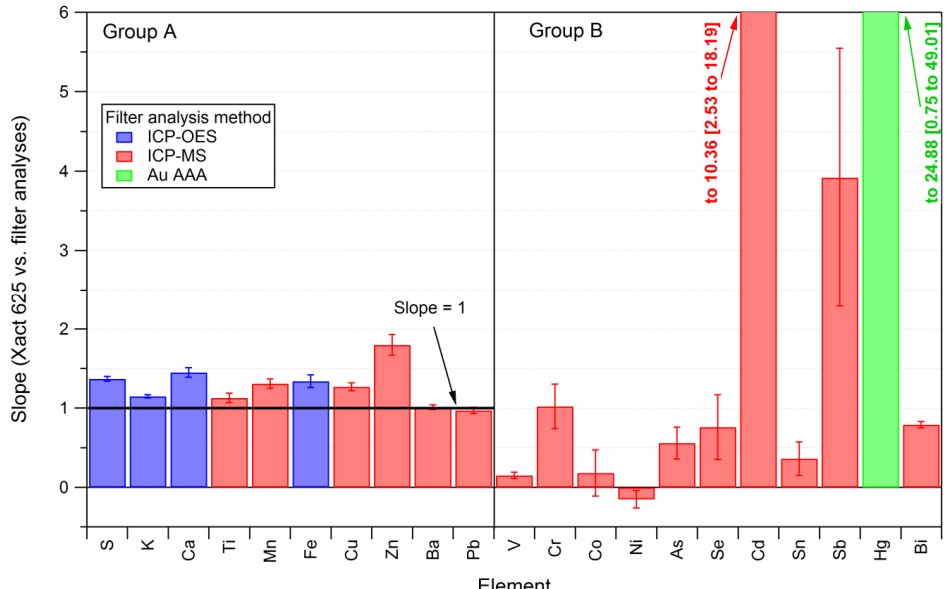

**Figure 3: Slopes of the Levenberg-Marquardt least squares fit regression analyses of Xact 625 vs. filter analyses with ICP-OES, ICP-MS and Hg AuAAA analysis. Error bars indicate the computed uncertainties of the fits.**

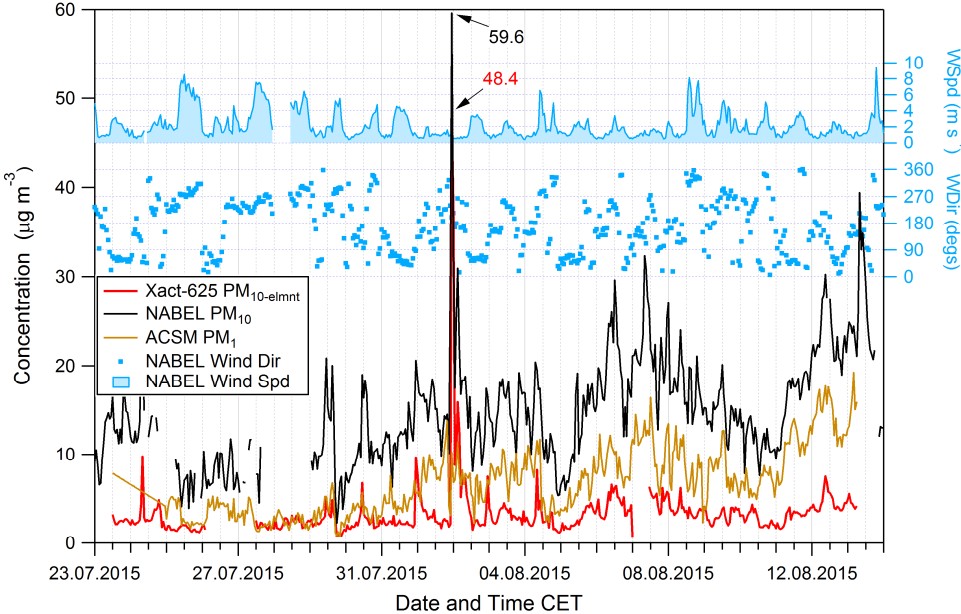

5 **Figure 4: Time series of Xact625 total elemental concentration (red), NABEL TEOM PM$_{10}$ data (black), and wind speed (WSpd) and direction (WDir) measurements (blue) in Härkingen.**



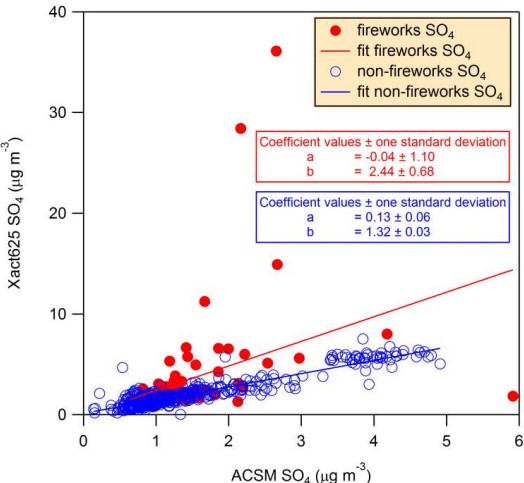

**Figure 5: Comparison of Xact PM$_{10}$ SO$_4$ vs. ACSM PM$_1$ SO$_4$. Data were split into fireworks (red) and non-fireworks (blue) periods.**

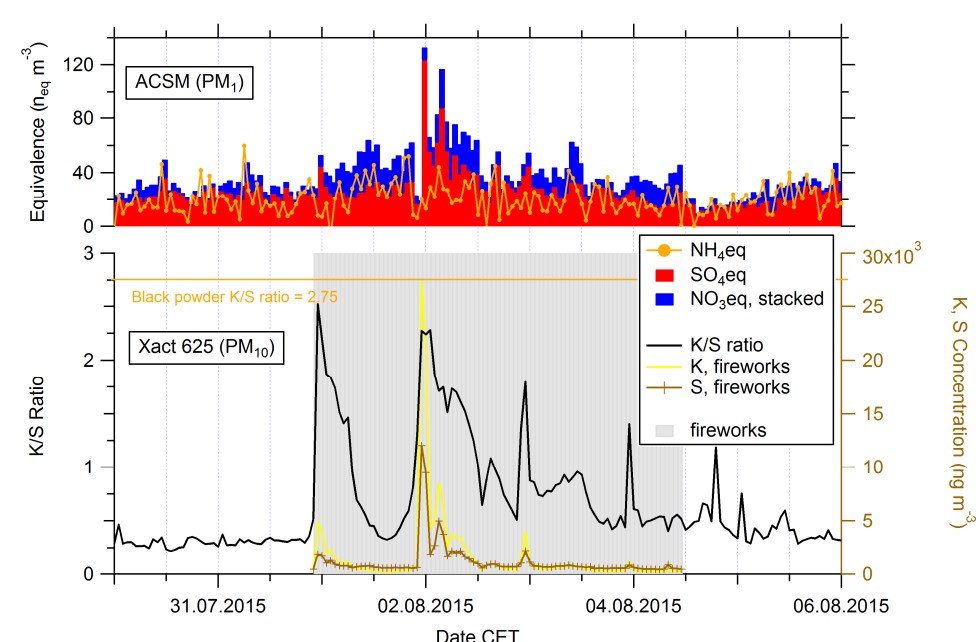

**Figure 6: Top: Time series of equivalent concentrations of NH$_4$, NO$_3$ and SO$_4$ in PM$_1$ measured with the ACSM. The NO$_3$eq is stacked on SO$_4$eq. Bottom: Time series of the K/S mass ratio (black, left axis) of the Xact 625. The K and S concentrations for the fireworks period (31 July 2200 CET to 4 Aug 1100 CET, shaded) are given in yellow and brown (right axis). The orange line**
10 **indicates the K/S mass ratio of 2.75 for black powder.**





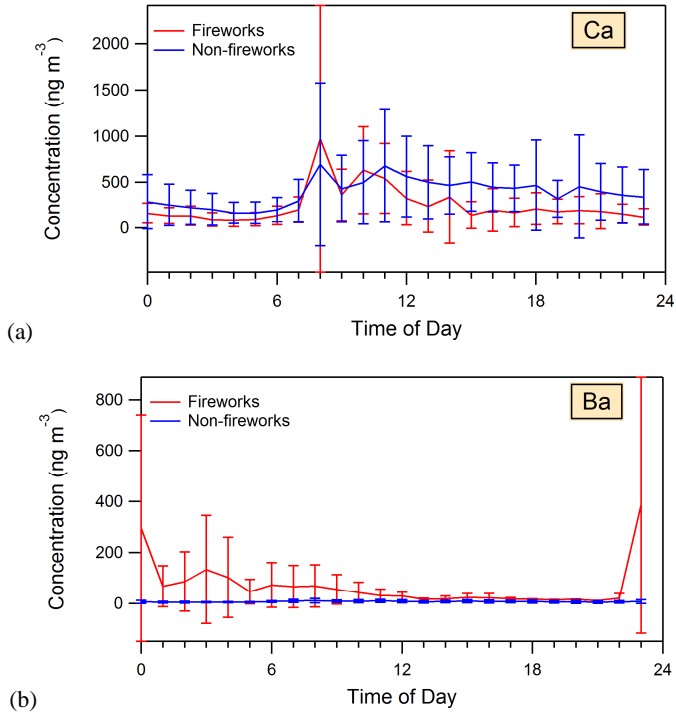

**Figure 7: Mean diurnal variations of a) Ca and b) Ba, stratified for fireworks (red) and non-fireworks (blue) periods. Error bars denote ± 1 standard deviation of the averaging period. Diurnal variations for the other elements are shown in the supplement S4 (Fig. S3).**

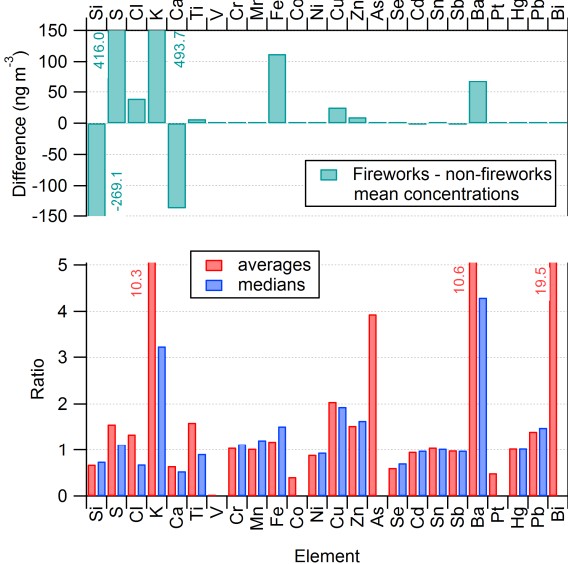

**Figure 8: Bottom: Enhancement ratios for all analysed elements for fireworks/non-fireworks classification. Top: 'Background'-subtracted mean concentrations of the south sector for the non-fireworks period. Numbers indicate values outside the axis range for Si (negative), S, and K.**





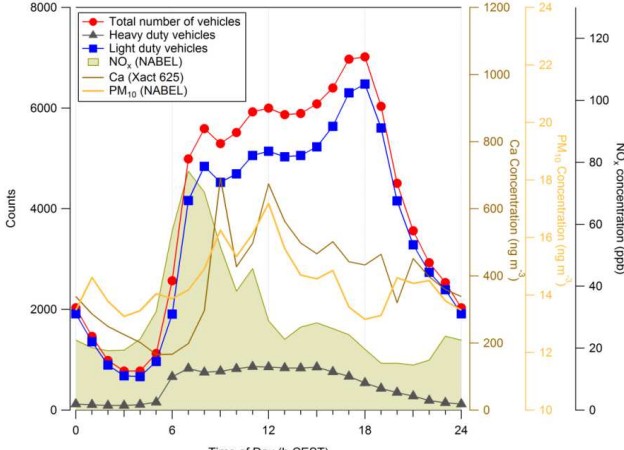

**Figure 9: Diurnal variation of traffic counts (average diurnal variations) and NO$_x$ concentration in Härkingen for the non-fireworks period. Note that the time axis is in CEST (Central European Summer Time = UTC + 2h = local time).**



(a)

(b)

(c)




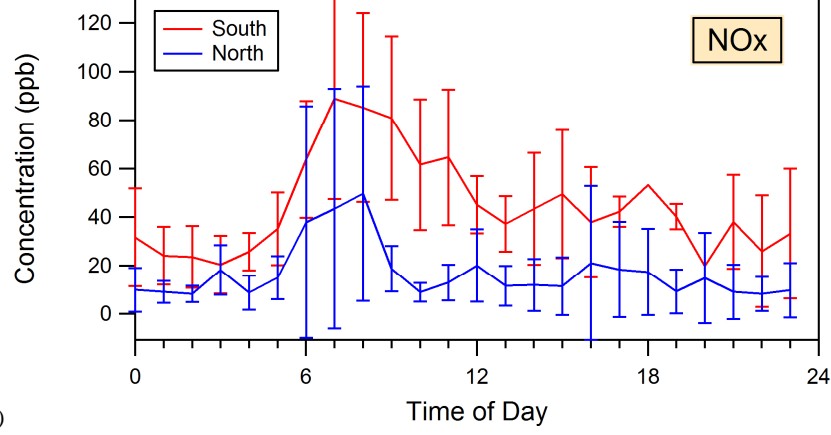

(d)

**Figure 10: Mean diurnal variations of a) S, b) Mn, c) Pb, and d) NO$_x$ for the non-fireworks period stratified for south (red) and north (blue) wind directions. Error bars denote ±1 standard deviation of the averaging period. Diurnal variations for the other elements are shown in the supplement S4 (Fig. S4).**

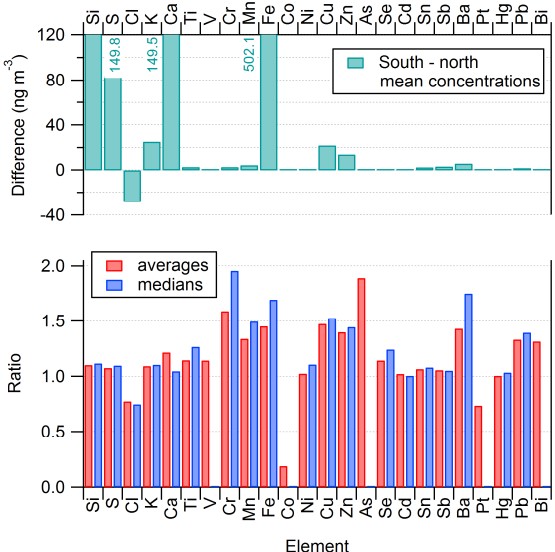

5   **Figure 11: Bottom: Enhancement ratios for all analysed elements for south/north sector classification for the non-fireworks period. Top: 'Background'-subtracted mean concentrations of the south sector for the non-fireworks period. Numbers indicate values outside the axis range for Si, Ca, and Fe.**





Atmospheric Measurement Techniques — Open Access — Discussions — EGU

## 10    Tables

**Table 1: Data characteristics of Xact 625 measurements in Härkingen, and minimum detection limits (MDL) for Xact and ICP/Hg AuAAA. Elements are sorted according to the groups in Table 2). Data were classified into fireworks and non-fireworks periods. The non-fireworks period was further classified into north (rural) and south (freeway) sectors according to the wind direction.**
5 **Numbers in italics indicate cases where the daily averages were <MDL. The cases for the two wind sectors do not add up to the non-fireworks cases as wind data are missing for a total of 12 h (cf. Fig. 1).**

| Element | Non-Fireworks avg | sdev | max | median | Fireworks avg | sdev | max | median | South sector (non-fireworks) avg | sdev | max | median | North sector (non-fireworks) avg | sdev | max | median | Xact MDL (60 min) | Xact Pts>MDL % | ICP MDL (24 h) | ICP Pts>MDL % |
|---|---|---|---|---|---|---|---|---|---|---|---|---|---|---|---|---|---|---|---|---|
| # cases | 370 | | | | 86 | | | | 174 | | | | 184 | | | | | | | |
| (units) | ng m⁻³ | ng m⁻³ | ng m⁻³ | ng m⁻³ | ng m⁻³ | ng m⁻³ | ng m⁻³ | ng m⁻³ | ng m⁻³ | ng m⁻³ | ng m⁻³ | ng m⁻³ | ng m⁻³ | ng m⁻³ | ng m⁻³ | ng m⁻³ | ng m⁻³ | % | ng m⁻³ | % |
| S | 739.28 | 524.59 | 2508.00 | 601.85 | 1155.32 | 1666.73 | 12034.00 | 677.15 | 799.62 | 520.55 | 2141.00 | 650.32 | 707.39 | 532.53 | 2508.00 | 529.50 | 4.20 | 100.00 | 7.662 | 100 |
| K | 161.00 | 56.58 | 484.09 | 152.81 | 1661.10 | 3854.66 | 27349.00 | 493.69 | 173.72 | 57.35 | 395.34 | 168.27 | 152.42 | 54.59 | 484.09 | 144.79 | 1.60 | 100.00 | 37.808 | 100 |
| Ca | 390.66 | 384.69 | 3211.00 | 262.79 | 253.21 | 389.74 | 3109.00 | 140.82 | 434.34 | 393.86 | 2166.00 | 271.75 | 362.94 | 381.77 | 3211.00 | 255.70 | 0.68 | 100.00 | 49.195 | 90 |
| Ti | 11.44 | 8.12 | 43.38 | 8.79 | 18.29 | 36.16 | 282.23 | 8.04 | 12.75 | 8.69 | 39.56 | 10.63 | 10.61 | 7.52 | 43.38 | 8.30 | 0.51 | 100.00 | 1.043 | 90* |
| Mn | 7.10 | 4.62 | 26.98 | 5.72 | 7.30 | 3.88 | 22.21 | 6.99 | 9.02 | 5.18 | 26.98 | 8.16 | 5.47 | 3.30 | 26.87 | 4.85 | 1.40 | 100.00 | 0.264 | 100 |
| Fe | 587.41 | 428.85 | 2338.00 | 460.08 | 699.55 | 385.97 | 1909.00 | 699.78 | 792.89 | 463.19 | 2338.00 | 708.17 | 404.84 | 301.70 | 1828.00 | 310.16 | 0.48 | 100.00 | 3.398 | 100 |
| Cu | 24.07 | 17.69 | 109.34 | 20.07 | 49.28 | 48.72 | 371.81 | 38.91 | 32.27 | 18.92 | 109.34 | 27.48 | 16.48 | 12.74 | 72.41 | 10.70 | 0.41 | 100.00 | 0.055 | 100 |
| Zn | 18.67 | 16.84 | 143.37 | 14.31 | 28.56 | 18.94 | 104.12 | 23.42 | 24.95 | 21.08 | 143.37 | 18.81 | 13.31 | 8.86 | 67.33 | 11.62 | 1.70 | 100.00 | 0.959 | 100 |
| Ba | 7.12 | 5.49 | 25.33 | 5.25 | 75.39 | 169.25 | 1127.00 | 22.58 | 9.31 | 6.03 | 25.33 | 8.02 | 5.22 | 4.18 | 22.94 | 3.81 | 0.39 | 98.20 | 0.819 | 100 |
| Pb | 2.96 | 3.89 | 41.07 | 1.99 | 4.17 | 3.02 | 15.30 | 2.96 | 3.86 | 5.22 | 41.07 | 2.63 | 2.26 | 1.77 | 12.50 | 1.73 | 0.52 | 2.00 | 0.216 | 100 |
| V | 0.06 | 0.15 | 1.22 | 0.00 | 0.00 | 0.00 | 0.00 | 0.00 | 0.06 | 0.15 | 1.22 | 0.00 | 0.06 | 0.16 | 0.92 | 0.00 | 0.57 | 74.60 | 0.026 | 100 |
| Cr | 2.40 | 2.30 | 12.96 | 1.75 | 2.51 | 2.22 | 9.23 | 1.98 | 2.52 | 2.52 | 12.96 | 2.60 | 1.44 | 1.61 | 8.47 | 0.79 | 0.40 | 0.44 | 0.614 | 40 |
| Co | 0.02 | 0.08 | 0.70 | 0.00 | 0.01 | 0.02 | 0.23 | 0.00 | 0.05 | 0.05 | 0.55 | 0.00 | 0.03 | 0.10 | 0.70 | 0.00 | 0.20 | 67.80 | 0.018 | 70 |
| Ni | 0.62 | 0.65 | 10.32 | 0.54 | 0.56 | 0.35 | 1.95 | 0.51 | 0.37 | 0.64 | 2.14 | 0.59 | 0.60 | 0.85 | 10.32 | 0.48 | 0.25 | 4.00 | 0.581 | 0 |
| As | 0.02 | 0.14 | 1.31 | 0.00 | 0.09 | 0.31 | 1.91 | 0.00 | 0.19 | 0.04 | 1.31 | 0.00 | 0.01 | 0.08 | 0.84 | 0.00 | | 38.00 | 0.026 | 100 |
| Se | 0.27 | 0.32 | 4.39 | 0.20 | 0.16 | 0.13 | 0.44 | 0.14 | 0.31 | 0.41 | 4.39 | 0.24 | 0.23 | 0.21 | 0.90 | 0.16 | | 12.70 | 0.015 | 100 |
| Cd | 6.75 | 3.61 | 23.62 | 6.25 | 6.49 | 3.14 | 21.33 | 6.14 | 6.94 | 3.79 | 23.62 | 6.27 | 6.68 | 3.49 | 20.63 | 6.26 | 10.30 | 85.00 | 0.028 | 10 |
| Sn | 20.79 | 7.82 | 55.34 | 19.57 | 21.78 | 8.27 | 54.28 | 20.10 | 21.66 | 7.80 | 55.28 | 20.85 | 20.08 | 7.74 | 55.34 | 18.98 | 13.30 | 94.00 | 0.028 | 100 |
| Sb | 31.31 | 11.22 | 111.88 | 29.80 | 30.97 | 10.42 | 67.96 | 29.38 | 32.98 | 11.49 | 77.63 | 31.08 | 29.79 | 10.79 | 111.88 | 28.76 | 16.00 | 86.80 | 0.026 | 100 |
| Hg | 0.63 | 0.25 | 1.49 | 0.61 | 0.64 | 0.18 | 1.31 | 0.63 | 0.62 | 0.24 | 1.23 | 0.62 | 0.64 | 0.25 | 1.49 | 0.62 | 0.34 | 7.20 | 0.001 | 40 |
| Bi | 0.07 | 0.12 | 0.70 | 0.00 | 1.27 | 3.82 | 23.47 | 0.15 | 0.08 | 0.12 | 0.49 | 0.00 | 0.05 | 0.12 | 0.70 | 0.00 | 0.43 | | 0.015 | 100 |
| Si | 839.20 | 398.20 | 3415.00 | 713.75 | 570.13 | 223.75 | 1758.00 | 532.28 | 902.57 | 433.21 | 2258.00 | 792.43 | 796.09 | 360.64 | 3415.00 | 694.15 | | | | |
| Cl | 113.70 | 200.20 | 969.80 | 26.44 | 153.07 | 578.02 | 4455.00 | 18.11 | 94.66 | 181.53 | 969.80 | 19.72 | 111.73 | 198.09 | 957.00 | 30.02 | 0.41 | 1.75 | | |
| *Pt* | *-0.05* | *0.21* | *0.66* | *0.03* | *0.03* | *0.07* | *0.34* | *0.00* | *0.04* | *0.10* | *0.66* | *0.00* | *0.06* | *0.11* | *0.64* | *0.00* | | | | |

*Numbers in italics indicate Xact daily averages < MDL.*
*both for ICP-OES, ICP-MS*
MDL for Hg Gold Amalgam Atomic Absorption



**Table 2. Regression coefficients for the comparison of Xact 625 and offline data. The 1-h values of the Xact 625 were averaged to 24-h values. Primed quantities are uncertainties.**

| Element | a | ± a' | b | ± b' | r² | Group |
|---------|--------|-------|-------|-------|------|-------|
| S | -169.73 | 30.57 | 1.37 | 0.03 | 1.00 | |
| K | 52.42 | 19.15 | 1.15 | 0.02 | 1.00 | |
| Ca | 13.87 | 17.91 | 1.45 | 0.06 | 0.99 | |
| Ti | 5.58 | 0.57 | 1.13 | 0.06 | 0.98 | |
| Mn | 1.72 | 0.28 | 1.31 | 0.06 | 0.99 | A |
| Fe | 93.05 | 35.80 | 1.34 | 0.08 | 0.97 | |
| Cu | 4.93 | 1.27 | 1.27 | 0.05 | 0.99 | |
| Zn | -5.56 | 2.18 | 1.80 | 0.13 | 0.96 | |
| Ba | -1.62 | 1.73 | 1.01 | 0.03 | 0.99 | |
| Pb | 0.32 | 0.17 | 0.97 | 0.04 | 0.99 | |
| V | -0.03 | 0.03 | 0.15 | 0.05 | 0.57 | |
| Cr | 2.13 | 0.26 | 1.02 | 0.28 | 0.63 | |
| Co | 0.01 | 0.02 | 0.18 | 0.29 | 0.05 | |
| Ni | -0.55 | 1.02 | -0.15 | 0.13 | 0.14 | |
| As | -0.09 | 0.06 | 0.56 | 0.20 | 0.50 | |
| Se | 0.00 | 0.15 | 0.76 | 0.41 | 0.30 | B |
| Cd | 6.49 | 0.20 | 10.36 | 7.83 | 0.18 | |
| Sn | 19.73 | 0.96 | 0.36 | 0.21 | 0.27 | |
| Sb | 22.17 | 3.58 | 3.92 | 1.63 | 0.42 | |
| Hg | 0.64 | 0.06 | 24.88 | 24.13 | 0.12 | |
| Bi | -0.16 | 0.06 | 0.79 | 0.04 | 0.98 | |
| Si | | | | | | |
| Cl | | | | | | C |
| Pt | | | | | | |
| Group A | average slope | | 1.28 | | | |
| Group A | standard deviation | | 0.24 | | | |

| | Elements analysed with ICP-OES |
|---|---|
| | Elements analysed with ICP-MS |
| | Element analysed with Au AAA |





**Table 3. Comparison of Xact data with published ICP data of other campaigns.**

| | Xact 625 averages | | Switzerland | | | | | | | | |
|---|---|---|---|---|---|---|---|---|---|---|---|
| | All days | Non-fireworks days | Belp | Geneva | Basel, summer | Payerne | Zürich | Zürich, summer | Payerne, summer | Payerne, summer | Härkingen NABEL |
| Reference | | | 1) | 2) | 3) | 4) | 5) | 6) | 6) | 7) | 8) |
| # cases, size | 22 | 17 | PM8 | PM10 | PM10 | PM10 | PM10 | PM10 | PM10 | PM10 | PM10 |
| Sampling period | 2015 | 2015 | 1985/86 | 1996/97 | 1997/98 | 1998/99 | 2008/09 | 2009 | 2009 | 2012 | 2015 |
| Unit | ng/m³ | ng/m³ | ng/m³ | ng/m³ | ng/m³ | ng/m³ | ng/m³ | ng/m³ | ng/m³ | ng/m³ | ng/m³ |
| Si | 783.14 | 829.45 | | | | | 210.90 | 571.00 | 634.00 | 370.00 | |
| S | 822.74 | 790.19 | | | | | 2394.30 | 625.00 | 637.00 | 360.00 | |
| Cl | 113.17 | 109.40 | | | 41.00 | | 656.50 | 66.00 | 190.00 | 30.00 | |
| K | 408.00 | 166.56 | | | 630.00 | 98.00 | 1318.20 | 187.00 | 188.00 | 120.00 | |
| Ca | 367.99 | 397.57 | | | 720.00 | 100.00 | 137.40 | 451.00 | 355.00 | 180.00 | |
| Ti | 12.26 | 11.30 | | | 38.00 | | 6.50 | 13.90 | 14.30 | 9.90 | |
| V | 0.05 | 0.06 | 3.90 | | | 0.70 | 0.40 | 1.00 | 1.20 | 0.70 | |
| Cr | 2.39 | 2.43 | | | 8.00 | | 0.90 | 2.30 | 1.60 | 1.00 | |
| Mn | 7.11 | 7.21 | 31.70 | | 16.00 | 2.80 | 5.80 | 7.20 | 5.00 | 2.80 | |
| Fe | 600.48 | 593.92 | | | 760.00 | 89.00 | 389.70 | 455.00 | 202.00 | 130.00 | |
| Co | 0.01 | 0.02 | | | | | | 2.60 | 0.10 | 0.10 | |
| Ni | 0.60 | 0.63 | | | 8.00 | 1.20 | 0.60 | 0.90 | 1.00 | 0.50 | 0.90 |
| Cu | 28.04 | 24.45 | 7.90 | 35.00 | 75.00 | 6.00 | 28.10 | 17.40 | 4.30 | 2.80 | 19.70 |
| Zn | 20.23 | 19.04 | 65.00 | 120.00 | 73.00 | | 20.30 | 16.10 | 9.40 | 7.50 | |
| As | 0.03 | 0.02 | 2.20 | 2.00 | 1.00 | 0.53 | | 0.30 | 0.50 | 0.20 | 0.31 |
| Se | 0.27 | 0.30 | | 6.00 | | 0.16 | | 0.30 | 0.30 | 0.20 | |
| Cd | 6.57 | 6.67 | 0.88 | 0.40 | 0.00 | 0.32 | | 0.10 | 0.10 | | 0.07 |
| Sn | 21.00 | 20.93 | | | | | 2.60 | 2.60 | 1.00 | | |
| Sb | 31.33 | 31.59 | | | 29.00 | 0.26 | 2.50 | 2.40 | 0.50 | | |
| Ba | 18.09 | 7.12 | | | 110.00 | | 6.70 | 6.50 | 3.90 | 1.80 | |
| Pt | 0.04 | 0.05 | | | | | | | | | |
| Hg | 0.64 | 0.64 | | 0.50 | | | | | | | |
| Pb | 3.13 | 3.04 | 134.00 | 95.00 | 51.00 | 10.00 | 14.20 | 3.60 | 3.10 | 1.20 | 4.90 |
| Bi | 0.26 | 0.07 | | | | | | 0.20 | 0.10 | | |

| 1) | Gälli et al. 1990 |
|---|---|
| 2) | Chiaradia and Cupelin 2000 - fortnight averages |
| 3) | Röösli et al. 2001 |
| 4) | Hueglin et al. 2005 |
| 5) | Richard et al. 2011 |
| 6) | Minguillón et al. 2012 |
| 7) | Alastuey et al. 2016 |
| 8) | BAFU/Empa 2015 - annual mean values |