# Peer review of "Elemental composition of ambient aerosols measured with high temporal resolution using an online XRF spectrometer"

_Atmospheric Measurement Techniques, 2016_

## Referee Comment (RC1) · Anonymous Referee #1 · 8 Feb 2017

The approach is generally sound and scientific. The abstract is too long and should be more focused on what the real novel results are. It reads more like a conclusion that an abstract. Figure 2 is too small and too many to be useful for interpretations. Needs to be redone with a focus on what the authors want us to see from this figure. Data in tables 1 and 2 are excessive and again what is it the authors really want is to observe from these tables. Maybe plots of MDLs versus elements would be a better way to see this and more efficient. Why are regression tables like Table 2 useful - maybe a few sentences in the text with a few selected plots would show these correlations better.

---

## Referee Comment (RC2) · Anonymous Referee #2 · 8 Feb 2017

The aim of the study is evaluating the operation and the data quality of an online XRF spectrometer (namely Xact 625, which allows obtaining 1-h time resolution elemental concentrations). Daily averaged elemental concentrations are compared with ICP analysis of daily samples. Results are of interest as the use of relatively simple online and high time resolution instrumentation for the measurement of PM composition may be very useful in many situations; and it is thus very important to verify the quality of these instruments.

The study is generally scientifically robust and well written. The comparison with ICP data is correctly carried out, showing both elements with good ICP-Xact agreements and elements that are not well quantified by the spectrometer. However, in my opinion,

some criticism (on both the use of this instrument and the comparison method) should be more explicitly quoted and discussed.

It is true that synchrotron-XRF or PIXE require expensive and not-easy-to-obtain accelerator time, but at the same time (if the experiment set up is properly optimized) these techniques allow a very accurate elemental analysis of an high number of samples collected in many sampling sites in very short times, while it is difficult to have many online spectrometer to simultaneously collect the PM in different locations.

For the elements of group A (main PM elements), it is true that the Xact-ICP correlation is very good, but it is not sufficient to say that their concentrations are well reproduced by the spectrometer (as stated in the conclusions and in the abstract). Intercepts arrive up to 40% of the average concentrations (as stated in the paper) and, even if they are not so big, I would not say that they are "small" (pag. 7) or negligible. Deviations of slopes from unit, although, again, not very big, are however significant (Xact/ICP ranging from 1 to 1.8, average 1.28). Possible reasons, like sampling and X-ray absorption, are suggested, but, as the authors themselves state, they are not completely supported/demonstrated by this study. Also they are not always convincing. In particular, X-ray absorption would produce underestimation while Xact concentrations are higher than those obtained by ICP; sampling would produce higher deviations for elements in big particles, while also S slope (1.37) significantly deviates from 1; slope of Zn (1.8) is significantly higher than the others. In this situation, it is not possible to conclude that the spectrometer correctly reproduces the concentrations of all elements of group A and that systematic differences have been attributed to specific reasons (as reported in the conclusion section). Looking at obtained results, I would conclude that correlation is very good (for group A), not big but significant differences are however observed (lower than. . . ), possible reasons have been investigated but further studies are needed.

(I think it is important to keep Table 2 and all the panels of Figure 2, but fonts should be bigger as it is difficult to read them as they now are).

It is also important noting that the comparison is made on daily averages and the accuracy of hourly concentrations has not been directly tested.

Finally, I have some comment on the description of the spectrometer (section 2.3). There are in my opinion important pieces of information that are not reported and that would be very useful: sampling area, irradiated area and X-ray detector used (including entrance window and collimation size). Minimum detection limits reported in Table 1 seam very small for 1-h sampling. Uncertainties are surely much higher that 5% for concentrations close to MDLs.

---

## Referee Comment (RC3) · Anonymous Referee #3 · 12 Feb 2017

The paper describes the outcomes of a field test to verify the performance of a quasi on-line XRF system (Xact 625) in the measurement of the elemental composition of PM. The data produced by the Xact system are compared with off-line standard analyses on PM collected on filters sampled by other samplers deployed in the same site. These kind of comparison are always interesting and I thing that the paper deserves the publication even if several corrections/improvements are necessary. I have how main issues and a series of punctual comments:

Issue 1: in the discussion of the level of agreement between the Xact 625 results with the other standard techniques, the Authors consider that some differences could be due the use of different sampling devices placed not exactly in the same position. My

question is: why the substrates used in the Xact625 have not been analyzed off-line by other techniques ? This would have removed any possible ambiguity related to different amount of sampled material...I have not found in the text any comment on this possibility. I consider a detailed discussion on this point absolutely necessary.

Issue2: the Authors discuss quite in deep the differences in the PM composition in the two periods (with and without fireworks) of the campaign. I do not find in such discussion any new or general element which could deserve to stay in the text. I think that this falls outside the main focus of the paper and that most of this discussion should be moved to the on-line supplementary material (or maybe it should find space in some local report). On this point, see the punctual comments below

Punctual comments: Abstract, line 17: the wording "Xact PM10 mass" could be misleading since by ED-XRF just a small fraction of the elements presents in PM10 can be detected. I recommend to use "the total concentration of the elements detected by Xact in PM10" Abstract, line 19: Begin the statement with "Ten" instead of "10" Introduction, line 38: replace "historically required" with "require" Introduction, line 39: This is not true: there are well known methods (e.g.: streaker sampler + PIXE, DRUM impactor + SXRF) which provide hourly or even sub-hourly time resolution with very low MDL. The statement must be changed accordingly. Page 2, line 3: replace "similar X ray facility" with "accelerator facilities" Page 2, line 5: delete "overwhelming" Page 2, line 10: the advantages offered by high time resolution have been discussed in literature well before the "older" reference given in the list...to my memory come some papers dating back to the eighties and I think that the Authors should be more precise on this point. Two reference papers are: Annegarn et al., Source profiles by unique ratios (SPUR) analysis: Determination of source profiles from receptor-site streaker samples. Atmos. Env. 26, 1992 D'Alessandro et al., Hourly elemental composition and sources identification of fine and coarse PM10 particulate matter in four Italian towns. JAS 34, 2002

Page 3, line 37-38: a list of element detectable by Xact is reported with the explanation

that the system sensitivity has been determined for each element by a reference sample. Actually, there are in the list couples of elements which interfere in a X-ray analysis (Fe-Co, Pb-As, Ba-Ti) and I really wonder that, for instance, Co and As can been safely detected in ambient aerosol (usually much richer in Fe and Pb). I have not found in the text any comment of this point. I think that the calibration procedure should be better described including a discussion on these possible interferences. This impacts on the data summarized in Table 1 too.

Page 4, first lines: the procedure to determine the MDL takes into account the spectrum collected in a blank portion of the filter. This ways the MDL get underestimated since, in each portion of the spectrum, the continuum is not due to the filter only but also to the tails of all the peaks due to the PM elements. Moreover, this method completely not consider the interferences discussed above. More realistic MDL values should be given for each element and a given sampling/analysis time in a dedicated table but should be calculated as an average of PM samples.

Page 8, Par 3.3: the whole section with related figures should move to the supplementary material Pag. 11, line 9-14: the origin of possible discrepancy have not been clearly identified and the wording "are attributed" is not correct and shold be replaced by "could be attributed"

Pag. 11 , line 25-31: these lines should be removed from the conclusions. . .they are of very local interest and more than a conclusion are just a summary of the findings Figure 1: the plot in the top panel should be shown in log. Scale: the present picture is not so informative Figure 2: the number of digits In the value of the a and b coeff. Is too high (i.e. show just significant digit). The values of the correlation coeff. Should be added in each plot. Figure 3: In my opinion the right part of the picture (from V on) does not give any information and should be deleted Figure 5: the fit of the red points is more or less meaningless. . ..please add the R2 values in the plot both for blue and red points Table 2: should be deleted by inserting the R2 values in the plot of fig. 2

---

## Author Response (AR1)

**Author comment: Additional corrections**

During the discussion stage of the manuscript we realized an issue with the time axes, which meanwhile could be completely resolved. The issue required corrections for those data where NABEL and Xact data were combined, i.e. for the North/South source apportionment. The regressions with the filter (ICP) data are not concerned, as these data were carefully (manually) extracted from the raw data. The changes concern Table 1 (new S1), Fig. 1 (right panel), Figs. 4, 11 (new 9), and S5, the time specifications in the text, and the mass budget estimate in Section 3.3. The changes to the statistics were minor and did not influence the conclusions. As some figures required corrections, we took advantage and also redrew other figures to make them consistent (axes ranges, element groups). Resulting changes to the text were applied where necessary.

The following changes were made:

All times are now reported in local time LT (= CET + 1 = UTC + 2).

Table 1 (new S1): Statistics for non-fireworks north and south sectors updated.

Table 3 (new 2): Column 2 ("all days") removed.

Fig. 1: Right panel updated.

Fig. 4: Xact data shifted by 1 h. The fireworks peaks of the Xact and the TEOM show a 1-h difference.

Fig. 8 (new 7): Elements arranged into groups A, B, C as in the text.

Fig. 11 (new 9): Data updated. Elements arranged into groups A, B, C as in the text.

The mass budget estimate had to be adjusted. Instead of comparing just the two 1-h maxima (TEOM and Xact), we integrated the mass over the full peaks (2 h for Xact and ACSM; 3 h for TEOM).

**Response to Referee #1**

(Referee's statements in *italics*)

The authors would like to thank the referee for careful reading and critical commenting of the manuscript.

*The abstract is too long and should be more focused on what the real novel results are. It reads more like a conclusion that an abstract.*

We shortened the abstract from 408 to 262 words, deleting the details referring to source apportionment. We focussed the abstract on the main findings regarding the intercomparison of the Xact with external measurements.

*Figure 2 is too small and too many to be useful for interpretations. Needs to be redone with a focus on what the authors want us to see from this figure.*

Figure 2 was redrawn, re-scaling the axes with respect to the maximum values of the filter data (ICP). Labels were enlarged. We included all data, spread over four panels, as we find it important to show the complete range of variability.

*Data in tables 1 and 2 are excessive and again what is it the authors really want is to observe from these tables. Maybe plots of MDLs versus elements would be a better way to see this and more efficient. Why are regression tables like Table 2 useful - maybe a few sentences in the text with a few selected plots would show these correlations better.*

We rearranged Tables 1 and 2. We moved the MDL information from Table 1 to Table 2. Then we moved Table 1, which now only comprises of statistical data characterizing the different periods, to the supplementary material (Table S1). Table 2 (new 1) now contains the regression coefficients and the MDLs. The data of Table 2 is presented in the new Figure 3, which shows the comparison between ICP and Xact MDLs, slopes and intercept-to-average concentration ratios for all studied elements. We consider it as advantageous when the data represented in Figures 2 and 3 can also be looked up quantitatively in a Table (as also supported by Referee #2), showing the full variability of slopes and intercepts.

**Response to Referee #2**

(Referee's statements in *italics*)

The authors would like to thank the referee for thoughtful reading and critical commenting of the manuscript.

*The comparison with ICP data is correctly carried out, showing both elements with good ICP-Xact agreements and elements that are not well quantified by the spectrometer. However, in my opinion, some criticism (on both the use of this instrument and the comparison method) should be more explicitly quoted and discussed.*

We add a statement on other possible sampling techniques and their benefit (see also remark further down).

We add a reference to Yatkin et al. (2012) and (2016).

"The NABEL network provided the reference for previous data intercomparisons (Hueglin et al., 2005; Lanz et al., 2010), as well as for the intercomparisons of this study. Comparisons between SR-XRF and filter samples analysed with ICP-OES and ICP-MS have been performed previously (Richard et al., 2010). Comparisons of XRF on samples collected on different substrates were performed, e.g., by Yatkin et al. (2012). A recent interlaboratory comparison of $PM_{10}$ filter analysis methods is presented in Yatkin et al. (2016), where XRF/PIXE and ICP methods were compared for several metrics. Some of these metrics are also applied in this study."

We added a remark on the regression analyses in Section 3.1:

"The regression intercepts were not forced to be zero to enable examination of potential differences in the measurement accuracy of each of the compared methods, e.g. blank subtraction. The slopes are more relevant and indicate biases between the methods. Orthogonal least squares regressions metrics were calculated which incorporates measurement errors in both quantities being compared. The slopes differed by less than 3.5 % between the two regression methods for the Group A elements. Ba and Pb achieved an almost perfect match with slopes around 1 and negligible intercepts. The other extreme is Zn with a slope of 1.8. Ti is another peculiar case with a slope of 1.13 and the largest intercept/average concentration ratio of 0.37. On average, the Xact 625 yielded approximately 28% higher elemental concentrations than ICP for the Group A elements."

We added the statement in the last paragraph of the conclusions:

"Xact streamlines near-real time monitoring of multi-metals despite not being as cost effective relative to conventional samplers that could be deployed in larger numbers at many sites simultaneously, or that could sample several size fractions at once, although their actual analysis costs (laboratories, accelerator facilities and staffing needs) are not considered here and they may surmount the instrument costs manifold."

*It is true that synchrotron-XRF or PIXE require expensive and not-easy-to-obtain accelerator time, but at the same time (if the experiment set up is properly optimized) these techniques allow a very accurate elemental analysis of an high number of samples collected in many sampling sites in very short times, while it is difficult to have many online spectrometer to simultaneously collect the PM in different locations.*

True. We have added two statements covering these aspects in the introduction:

"Access restrictions limit the number of collected samples to be analysed, and hence field campaigns are predominantly episodic."

"Instruments of this type can be used for continuous (months, years) monitoring at a site, but their cost restricts the simultaneous deployment of a larger number of devices for different size fractions or at different sites. The benefit of long-term, quasi real time data access, favourable, e.g., for air quality monitoring, contrasts with the possibilities of relatively low cost, multi-site and multi-size samplers used so far in episodic field studies."

*For the elements of group A (main PM elements), it is true that the Xact-ICP correlation is very good, but it is not sufficient to say that their concentrations are well reproduced by the spectrometer (as stated in the conclusions and in the abstract). Intercepts arrive up to 40% of the average concentrations (as stated in the paper) and, even if they are not so big, I would not say that they are "small" (pag. 7) or negligible. Deviations of slopes from unit, although, again, not very big, are however significant (Xact/ICP ranging from 1 to 1.8, average 1.28). Possible reasons, like sampling and X-ray absorption, are suggested, but, as the authors themselves state, they are not completely supported/demonstrated by this study. Also they are not always convincing. In particular, X-ray absorption would produce underestimation while Xact concentrations are higher than those obtained by ICP; sampling would produce higher deviations for elements in big particles, while also S slope (1.37) significantly deviates from 1; slope of Zn (1.8) is significantly higher than the others. In this situation, it is not possible to conclude that the spectrometer correctly reproduces the concentrations of all elements of group A and that systematic differences have been attributed to specific reasons (as reported in the conclusion section). Looking at obtained results, I would conclude that correlation is very good (for group A), not big but significant differences are however observed (lower than. . . ), possible reasons have been investigated but further studies are needed.*

Agreed. We have changed our concluding statement to:

"Excellent correlation between Xact 625 and ICP-OES/ICP-MS was observed for 24-h averages of the elements S, K, Ca, Ti, Mn, Fe, Cu, Zn, Ba, and Pb ("Group A"). The daily averages calculated from hourly measurements by Xact were on average 30 % higher (range -3 % to +80 %, dependent on elements) than 24-h integrated filter measurements by ICP.   …   Further research on these issues is needed."

*(I think it is important to keep Table 2 and all the panels of Figure 2, but fonts should be bigger as it is difficult to read them as they now are).*

We have rearranged Table 1 by moving the MDL part to Table 2 and the remainder of Table 1 to the supplement which now only contains statistics of the complete Xact dataset. We have also redrawn Figure 2 completely and have replaced Figure 3 with a new one showing the interference free MDLs of Xact and ICP MDLs as well as the slopes (with standard deviations) and the ratios of the intercept with the average concentrations for each element. In this way, complete information on all intercomparison data is presented in graphical and tabular form for quick reference. See response to referee #2.

*It is also important noting that the comparison is made on daily averages and the accuracy of hourly concentrations has not been directly tested.*

It is true that the hourly concentrations have not been compared to ICP data at their original time resolution, since the ICP data are only available with 24h time resolution, and we included text in the manuscript stating this limitation. Nevertheless, while indeed the emphasis is on the intercomparison of 24-h averages of filters and Xact, hourly S data from Xact have been compared with $PM_1$ $SO_4$ collected with the ACSM (Figure 5). On the other hand, the estimated mass budget in Section 3.3 is also based on hourly data (2 h interval for the Xact, 3 h interval for the TEOM, to capture the full peak period with high concentrations). Fig. 4 shows hourly resolved time series of Xact, TEOM and ACSM data.

We insert the following statement at line 29 of Section 2.6:

"Comparisons of hourly Xact data were only possible for S with the ACSM data (in the form of $PM_1$ sulphate, assuming that all S occurs in $PM_1$), and between the total Xact element mass and $PM_{10}$ of the NABEL TEOM instrument, see Sections 3.2 and 3.3."

And in the conclusions we added:

 "The accuracy of hourly values has only been tested for the case of the fireworks peak late on 1 August 2015, where the sum of all elements has been compared to the total mass of the NABEL TEOM. Good agreement between the Xact and TEOM mass was found when corrections derived from the 24-h filter analyses were applied. This was a special case dominated by just three elements, S, K, Cl, and a generalization to all measured elements is not recommended."

*Finally, I have some comment on the description of the spectrometer (section 2.3). There are in my opinion important pieces of information that are not reported and that would be very useful: sampling area, irradiated area and X-ray detector used (including entrance window and collimation size).*

True. However, having such proprietary information published is of severe concern for CES due to potential competition with other manufactures. Furthermore, we do believe that removal of such information is consistent with discussion of other methodologies (for example, most journal papers focused on ambient data from say, ICP-MS do not provide details of the liquid impingers or cones being used). We checked to ensure that the data interpretation is indeed not impacted by elimination of such details.

*Minimum detection limits reported in Table 1 seem very small for 1-h sampling.*

The reported MDLs are interference-free MDLs at the 1-sigma confidence level for unsampled tape (see reference Currie 1977 in the manuscript). These MDLs give an estimate on the minimum detectable mass for pure elements, i.e. for single element standards, hence they represent the achievable minimum of detection under ideal conditions. MDLs in a multi-element sample are expected to be higher, depending on sample composition and concentration, but they would have to be reported individually for each measurement. No such routine is presently implemented in the Xact. MDLs decrease with increasing sampling time ($\sim t^{-1/2}$).

"XRF based MDLs are inversely proportional to the square root of the X-ray analysis time (Currie, 1977), which in the case of Xact is same as the sampling duration. Hence, Xact MDLs are lower for longer sampling durations. Interference free MDLs, while true are idealized lower limits of detection of one single element. As with most analytical methods, matrix effects in ambient samples from interferences between different elements and analyte concentrations could potentially result in MDLs of ambient samples to be higher and vary across samples, which makes them difficult to generalize and report. It is therefore often preferred to report measurement uncertainties to characterize a measurement. "

*Uncertainties are surely much higher that 5% for concentrations close to MDLs.*

Agreed. We changed the wording from 'may be higher' to 'are higher' (p 4, line 14):

An uncertainty of 5 % or less due to fitting errors and uncertainties in the standards has been derived from laboratory experiments with NIST standards (benchtop XRF, filter analyses). Uncertainties are expected to be higher for concentrations close to the MDL; for elements with potential for line interferences in multi-element samples; and, from self absorption effects for lightest elements (Si, S, Cl, K, Ca).

**Response to Referee #3**

(Referee's statements in *italics*)

The authors would like to thank the referee for thoughtful reading and critical commenting of the manuscript.

*Issue 1: in the discussion of the level of agreement between the Xact 625 results with the other standard techniques, the Authors consider that some differences could be due the use of different sampling devices placed not exactly in the same position. My question is: why the substrates used in the Xact625 have not been analyzed off-line by other techniques? This would have removed any possible ambiguity related to different amount of sampled material...I have not found in the text any comment on this possibility. I consider a detailed discussion on this point absolutely necessary.*

Good point. An offline analysis of the Xact sampling tape was not considered for two reasons: 1) Hourly samples do not collect sufficient mass for ICP analysis. 2) The samples collected are typically not amenable to post-sampling offline analysis due to potential for cross-contamination from sampled filter tape wound upon itself onto the filter wheel.

We added the following statement to the text in Section 2.3:

"While this approach is non-destructive, the samples collected are typically not amenable to offline analysis post-sampling due to potential for cross-contamination from sampled filter tape wound upon itself onto the filter wheel."

*Issue2: the Authors discuss quite in deep the differences in the PM composition in the two periods (with and without fireworks) of the campaign. I do not find in such discussion any new or general element which could deserve to stay in the text. I think that this falls outside the main focus of the paper and that most of this discussion should be moved to the on-line supplementary material (or maybe it should find space in some local report). On this point, see the punctual comments below.*

Partly agreed. It was originally the goal to demonstrate the capabilities of the Xact in field research in Switzerland, and to show an application of Xact data for source apportionment. From all three reviewers' comments we see that focussing on the first aspect was more desired than emphasizing the source apportionment. We therefore decided to shorten the source apportionment part, and to save part of the material for an upcoming publication. On the other hand, we elaborated the intercomparison part more deeply. This led to some re-writing of various parts of the manuscript, dropping two figures (Figs. 6 and 9), moving Table 1 to the supplementary material, and keeping the source apportionment part at a level demonstrating a few basic capabilities of the Xact data. We do, however, not agree with removing the whole Section 3.3 to the supplementary material, as it demonstrates important findings for operating the Xact. We instead condensed the section to the necessary, documenting the value of 1-h time resolution and the large dynamic range of the Xact (a concentration jump to 48 µg m$^{-3}$ within an hour). A comparison of the Xact's and TEOM's peaks could demonstrate how closely the Xact elemental mass represents the total measured PM$_{10}$ mass in this particular case. We consider this to be a significant finding of this study.

*Punctual comments:*

*Abstract, line 17: the wording "Xact PM10 mass" could be misleading since by ED-XRF just a small fraction of the elements presents in PM10 can be detected. I recommend to use "the total concentration of the elements detected by Xact in PM10"*

Agreed. However, when condensing the abstract as suggested by Referee #1 this sentence was deleted from the abstract.

*Abstract, line 19: Begin the statement with "Ten" instead of "10"*

Done.

*Introduction, line 38: replace "historically required" with "require"*

Done.

*Introduction, line 39: This is not true: there are well known methods (e.g.: streaker sampler + PIXE, DRUM impactor + SXRF) which provide hourly or even sub-hourly time resolution with very low MDL. The statement must be changed accordingly.*

Agreed. We added the sentences and references at the appropriate place:

"For high time resolution, impactors are used where the sample is collected on a foil (e.g. a rotating drum impactor, Lundgren, 1967), or on a combination of an impactor plate and a filter, such as in a streaker sampler (e.g. Annegarn et al., 1992). These samples are then exposed to X-rays or a particle beam without further treatment, which provide quantitative data with low detection limits."

*Page 2, line 3: replace "similar X ray facility" with "accelerator facilities"*

Done.

*Page 2, line 5: delete "overwhelming"*

*Done.*

*Page 2, line 10: the advantages offered by high time resolution have been discussed in literature well before the "older" reference given in the list. . .to my memory come some papers dating back to the eighties and I think that the Authors should be more precise on this point. Two reference papers are: Annegarn et al., Source profiles by unique ratios (SPUR) analysis: Determination of source profiles from receptor-site streaker samples. Atmos. Env. 26, 1992 D'Alessandro et al., Hourly elemental composition and sources identification of fine and coarse PM10 particulate matter in four Italian towns. JAS 34, 2002*

Agreed. We added the two references and quoted them in the appropriate places in the introduction.

*Page 3, line 37-38: a list of element detectable by Xact is reported with the explanation that the system sensitivity has been determined for each element by a reference sample. Actually, there are in the list couples of elements which interfere in a X-ray analysis (Fe-Co, Pb-As, Ba-Ti) and I really wonder that, for instance, Co and As can been safely detected in ambient aerosol (usually much richer in Fe and Pb). I have not found in the text any comment of this point. I think that the calibration procedure should be better described including a discussion on these possible interferences. This impacts on the data summarized in Table 1 too.*

Very good point. We added some text discussing this point:

"Line interference is well-known for element couples like Fe-Co, Pb-As, Ba-Ti and makes detection of one element difficult if the other is abundant in the sample. The linear least squares reference deconvolution algorithm implemented in the Xact fits the measured sample spectrum with the library of pure element reference spectra to resolve concentrations of each calibrated element."

The last remark on the impact of line interference on the data in Table 1 concerns only the pair Ba-Ti, as for each other pair one element is below MDL. We found a high correlation between Ti and Ba for the fireworks ($r^2$=0.94), where it is expected, and a different, but still good correlation ($r^2$= 0.55) for the non-fireworks cases (see attached Figure), where both elements also show a different behaviour and different enhancement ratios (Fig. 7 in the manuscript). This is a hint towards a good element separation despite the potential Ba-Ti line interference. In addition, from Fig. 2 it can be seen that both Ba and Ti are highly correlated with their corresponding ICP-MS data, with the largest values corresponding to the fireworks cases. One would expect

the Ti values of the fireworks to deviate more from the ICP data when Ba interference were not adequately considered.

We added the following sentences:

"Potential line interference between Ti and Ba can be largely exluded, because the element couple reveals two different regressions for fireworks and non-fireworks cases, as well as distinct diurnal variations in the non-fireworks cases."

"The element couples of Fe-Co and Pb-As do not show correlations within the couples, because most of their data points are below their respective MDLs, and no conclusion about the deconvolution of interfering lines can be drawn for these elements. Comparing the Xact values with the NABEL annual mean values (Table 2) shows differences smaller than 40 % for Cu, Pb, and Ni, while the differences are much larger for As and Cd. The latter two elements are below their respective MDL (Fig. 2)."

*Page 4, first lines: the procedure to determine the MDL takes into account the spectrum collected in a blank portion of the filter. This ways the MDL get underestimated since, in each portion of the spectrum, the continuum is not due to the filter only but also to the tails of all the peaks due to the PM elements. Moreover, this method completely not consider the interferences discussed above. More realistic MDL values should be given for each element and a given sampling/analysis time in a dedicated table but should be calculated as an average of PM samples.*

Partly agreed. The MDLs reported by the manufacturer are interference-free, 1-sigma confidence level, minimum detection limits determined for single element standards. They are clearly defined and well reproducible, and they are re-determined at regular intervals (1 to 3 months). They indicate the lowest possible detection limits under ideal conditions. While we agree that the table suggested by Referee #3 would be very helpful and probably more meaningful, its determination would require numerous experiments with different sample compositions and concentrations which is beyond the scope of the current study.

"Interference free MDLs, while true are idealized lower limits of detection of one single element. As with most analytical methods, matrix effects in ambient samples from interferences between different elements and analyte concentrations could potentially result in MDLs of ambient samples to be higher and vary across samples, which makes them difficult to generalize and report."

*Page 8, Par 3.3: the whole section with related figures should move to the supplementary material*

Partly agreed, see issue 2, above. We condensed the Section 3.3 to the necessary, but want to keep it in place. We consider the Section relevant for the article.

We try to clarify this point with the following text sequence:

"Investigation of the highest peaks reveals the performance of the Xact under high load conditions, when sample thickness may become critical for XRF analysis. A comparison of the two instruments' peaks could demonstrate how closely the Xact mass represents the total measured PM10 mass. Inspection of the different time series indicates that the TEOM peak is broader (3 h) and higher (59.6 µg m-3), and its maximum concentration is reached 1 h later (at 2 Aug 2015 0000 LT), but its increase in concentration starts at the same time as the Xact (at 2200 LT). The delay in the maximum can be attributed to the time constant of the TEOM used for reducing measurement noise and to the averaging procedure. For a comparison of the two peaks their measured masses were integrated over the duration of the peaks, i.e. over 2 h for the Xact data and the ACSM data, and 3 h for the NABEL data.."

*Pag. 11, line 9-14: the origin of possible discrepancy have not been clearly identified and the wording "are attributed" is not correct and should be replaced by "could be attributed"*

Done.

*Pag. 11 , line 25-31: these lines should be removed from the conclusions. . .they are of very local interest and more than a conclusion are just a summary of the findings*

Agreed. Done.

*Figure 1: the plot in the top panel should be shown in log. Scale: the present picture is not so informative*

Not agreed. We want to emphasize the size of the fireworks peak in relation to the background concentrations, and this is best visible with a linear y-axis. Furthermore, on a logarithmic scale with stacked element concentration, the first element is shown with proportionally more area than the subsequent elements, yielding a distorted picture of its importance. Therefore, we prefer to leave the linear scale on the y-axis in the top panel as is.

*Figure 2: the number of digits In the value of the a and b coeff.  Is too high (i.e. show just significant digit). The values of the correlation coeff. Should be added in each plot.*

Agreed. However, we redrew Fig. 2 completely and moved all the regression coefficients to (new) Table 1 (former Table 2), where all coefficients are given to 2 decimal digits. The new figure represents the elements in Group A with colours, while Group B and C elements are represented in black and white, indicating that their data has issues with MDLs and others. Still, we retain all data in the graph to present the complete picture – in accordance with referee #2. We have also reordered that data in Figs. 7 and 11 (new: 7 and 9) according to the groups, hence we consistently show good and bad data throughout the article.

*Figure 3: In my opinion the right part of the picture (from V on) does not give any information and should be deleted*

Partly agreed. We completely redrew Fig. 3, adding the MDLs for Xact and ICP, the regression slopes and intercept-to-average ratios for all elements. See also comment on Fig. 2, above.

*Figure 5: the fit of the red points is more or less meaningless. please add the $R^2$ values in the plot both for blue and red points*

Agreed. Done.

*Table 2: should be deleted by inserting the $R^2$ values in the plot of fig. 2*

The changes made to Figs. 2 and 3 and Table 2 make this suggestion obsolete.

[revised manuscript text omitted]
 19 %. Similarly the agreement between each of the labs and bench top XRF is good as well. If Se and Ca are excluded the average percent difference between XRF and IDEA is 5.4% while the difference between XRF and ERG is -3.1%. The comparison of the daily averaged Xact values with the benchtop XRF values shows an average difference of 37 % (Xact-CES)/CES) for the elements Zn, Cu, Fe, K, Ca, and Mn, which is close to the observed mean difference to ICP. It is also consistent in the sense that all average differences Xact – CES for these elements are positive. The benchtop XRF and the Xact are typically within 5% when analysing the same standard. Further both benchtop XRF and Xact use the same type of fitting routine (with minor differences in the determination of spectral background), hence the most likely explanation for the difference between the Xact and the labs is differences due to sampling or sampling location.

**S3. Spiked filter samples for method intercomparisons**

CES produced a set of six quartz filters coated with known amounts of the elements Zn, Sr, Cu, Pb, and Fe. These filters were analysed with a benchtop XRF instrument by CES, and three each of them were sent to IDAEA-CSIC, and ERG for analysis with ICP-MS. The results are presented in Table S2. Notice that Pb is not reported for XRF, because of large variations of the measured values for quartz filters. This indicates a problem with the XRF fitting routine for quartz filters, as the issue is not seen with Teflon filters.

**Table S3. Spiked filter analyses for five elements. Comparison between XRF and ICP-MS analyses performed at three independent laboratories.**

| Sample Start Time | Sample | Element | CES Spiked Conc. (ng cm⁻²) | XRF Conc. (ng cm⁻²) | Blank (ng cm⁻²) | IDAEA Conc. (ng cm⁻²) | ERG ERG Values (ng cm⁻²) | Blank (ng cm⁻²) | % Difference Spiked vs. CES (CES - spiked) / spiked | Spiked vs. IDAEA (IDAEA - spiked) / spiked | Spiked vs. ERG (ERG - spiked) / spiked | IDAEA vs. CES (CES - IDAEA) / IDAEA | ERG vs. CES (CES - ERG) / ERG | Average Percent Difference Spiked vs. CES (CES - spiked) / spiked | Spiked vs. IDAEA (IDAEA - spiked) / spiked | Spiked vs. ERG (ERG - spiked) / spiked | IDAEA vs. CES (CES - IDAEA) / IDAEA | ERG vs. CES (CES - ERG) / ERG |
|---|---|---|---|---|---|---|---|---|---|---|---|---|---|---|---|---|---|---|
| 21.04.2016 11:50 | PQ042116A | Zn | 97.4 | 88.1 | 10.2 | 133.1 | | | -9.6 | 36.6 | | -33.8 | | | | | | |
| 21.04.2016 12:25 | PQ042116B | | 97.4 | 89.2 | 10.2 | 155.0 | | | -8.4 | 59.1 | | -42.4 | | | | | | |
| 21.04.2016 13:00 | PQ042116C | | 97.4 | 83.5 | 10.2 | 97.7 | | | -14.3 | 0.3 | | -14.5 | | -9.7 | 32.0 | 22.4 | -30.3 | -24.6 |
| 21.04.2016 15:13 | PQ042116D | | 97.4 | 88.3 | 10.2 | | 104.1 | 30.7 | -9.3 | | 6.8 | | -15.1 | | | | | |
| 21.04.2016 15:46 | PQ012116E | | 97.4 | 92.4 | 10.2 | | 123.2 | 30.7 | -5.1 | | 26.4 | | -25.0 | | | | | |
| 21.04.2016 16:19 | PQ042116F | | 101.5 | 90.2 | 10.2 | | 136.0 | 30.7 | -11.2 | | 34.0 | | -33.7 | | | | | |
| 21.04.2016 11:50 | PQ042116A | Sr | 206.0 | 191.6 | | 178.6 | | | -7.0 | -13.3 | | 7.3 | | | | | | |
| 21.04.2016 12:25 | PQ042116B | | 206.0 | 194.0 | | 194.5 | | | -5.8 | -5.6 | | -0.2 | | | | | | |
| 21.04.2016 13:00 | PQ042116C | | 206.0 | 193.9 | | 147.8 | | | -5.9 | -28.2 | | 31.1 | | -6.2 | -15.7 | -7.0 | 12.7 | 0.8 |
| 21.04.2016 15:13 | PQ042116D | | 206.0 | 191.6 | | | 190.5 | 1.5 | -7.0 | | -7.5 | | 0.5 | | | | | |
| 21.04.2016 15:46 | PQ012116E | | 206.0 | 194.0 | | | 189.8 | 1.5 | -5.8 | | -7.9 | | 2.2 | | | | | |
| 21.04.2016 16:19 | PQ042116F | | 206.0 | 193.9 | | | 194.6 | 1.5 | -5.9 | | -5.5 | | -0.3 | | | | | |
| 21.04.2016 11:50 | PQ042116A | Cu | 127.6 | 108.1 | | 111.9 | | | -15.3 | -12.3 | | -3.4 | | | | | | |
| 21.04.2016 12:25 | PQ042116B | | 127.6 | 110.6 | | 117.8 | | | -13.3 | -7.7 | | -6.1 | | | | | | |
| 21.04.2016 13:00 | PQ042116C | | 127.6 | 111.9 | | 87.9 | | | -12.3 | -31.1 | | 27.2 | | -13.6 | -17.0 | 5.8 | 5.9 | -18.2 |
| 21.04.2016 15:13 | PQ042116D | | 127.6 | 108.1 | 0.8 | | 131.2 | 1.3 | -15.3 | | 2.8 | | -17.6 | | | | | |
| 21.04.2016 15:46 | PQ012116E | | 127.6 | 110.6 | 0.8 | | 129.4 | 1.3 | -13.3 | | 1.4 | | -14.5 | | | | | |
| 21.04.2016 16:19 | PQ042116F | | 127.6 | 111.9 | 0.8 | | 144.3 | 1.3 | -12.3 | | 13.1 | | -22.5 | | | | | |
| 21.04.2016 11:50 | PQ042116A | Pb | 20.5 | NR | | 22.6 | | | NR | 10.3 | | NR | | | | | | |
| 21.04.2016 12:25 | PQ042116B | | 20.5 | NR | | 37.9 | | | NR | 84.7 | | NR | | | | | | |
| 21.04.2016 13:00 | PQ042116C | | 20.5 | NR | | 20.9 | | | NR | 2.1 | | NR | | NR | 32.4 | 57.7 | NR | NR |
| 21.04.2016 15:13 | PQ042116D | | 20.5 | NR | | | 27.1 | 1.1 | NR | | 32.0 | | NR | | | | | |
| 21.04.2016 15:46 | PQ012116E | | 20.5 | NR | | | 29.1 | 1.1 | NR | | 41.8 | | NR | | | | | |
| 21.04.2016 16:19 | PQ042116F | | 20.5 | NR | | | 40.9 | 1.1 | NR | | 99.4 | | NR | | | | | |
| 21.04.2016 11:50 | PQ042116A | Fe | 3024.6 | 2759.6 | | 2827.6 | | | -8.8 | -6.5 | | -2.4 | | | | | | |
| 21.04.2016 12:25 | PQ042116B | | 3024.6 | 2795.5 | | 3543.9 | | | -7.6 | 17.2 | | -21.1 | | | | | | |
| 21.04.2016 13:00 | PQ042116C | | 3024.6 | 2786.4 | | 2377.9 | | | -7.9 | -21.4 | | 17.2 | | -8.1 | -3.6 | -0.8 | -2.1 | -7.2 |
| 21.04.2016 15:13 | PQ042116D | | 3024.6 | 2759.6 | 154.5 | | 2901.2 | 140.5 | -8.8 | | -4.1 | | -4.9 | | | | | |
| 21.04.2016 15:46 | PQ012116E | | 3024.6 | 2795.5 | 154.5 | | 2901.2 | 140.5 | -7.6 | | -4.1 | | -3.6 | | | | | |
| 21.04.2016 16:19 | PQ042116F | | 3024.6 | 2786.4 | 154.5 | | 3202.9 | 140.5 | -7.9 | | 5.9 | | -13.0 | | | | | |

Tests with specifically produced reference samples of Fe, Cu, Zn, Sr, and Pb (Table S2) showed relative differences between the measured concentrations and the theoretically expected concentrations ranging from -6.2 % (Sr) to -13.6 % (Cu) for benchtop XRF, on average -9.4 % (without Pb). For all these elements, XRF underestimated the expected value, -as expected for absorption of fluorescence radiation by the quartz fiber material (Tanner et al. 1974). Similarly spiked teflon filters (not shown) also showed underestimation of the expected concentrations, though not as much as for the quartz filters. A statistical analysis revealed that at the 99 % confidence level only Cu showed a significant difference between the two filter types. ICP showed differences between -17 % and +32 % (average 5.6 %) for IDAEA-CSIC, and -7 % and +58 % (average 15.6 %) for ERG for quartz filters. The scatter is much larger than for the field samples, and differences can be positive or negative.

**S3S4. Diurnal variations of elements for fireworks and non-fireworks periods**

[Figure]

[Figure]

[Figure]

[Figure]

[Figure]

**Figure S4: Diurnal variations of the Group A elements Si, S, Cl, K, Ti, Mn, Fe, Cu, Zn, and Pb. See Fig. 6.**

[Figure]

[Figure]

[Figure]

[Figure]

**S4S5.** Diurnal variations of elements for north and south wind sectors

**Figure S4S5: Diurnal variations of the Group A elements Si, Cl,  Ca, Ti,  Fe, Cu, Zn, _and_ Ba. South means a wind from the freeway towards the station. See Fig. 8.**

---

## Author Response (AR2)

**Response to the Associate Editor**

For both the main text and the Supplement:

The numeric concentration data and the percentage concentrations are given with too many significant figures. Three significant figures suffice when the first significant figure is a "1" and two significant suffice in the other cases.

Tables 1, 2, S1, S2, and S3 were modified according to the suggestion. We also changed the values in the text (Section 3.3) accordingly.

For the main text:

| | |
|---|---|
| Page 1, line 29: Replace "proton-induced" by "particle-induced". | Done. |
| Page 1, line 30: Replace "i.e. sample" by "i.e., sample". | Done. |
| Page 1, line 33: Replace "e.g. a" by "e.g., a". | Done. |
| Page 1, line 34: Replace "e.g. Annegarn" by "e.g., Annegarn". | Done. |
| Page 1, line 36: Replace "XRF" by "The XRF". | Done. |
| Page 2, line 11: Replace "e.g. resuspension" by "e.g., resuspension". | Done. |
| Page 2, line 17: Replace "i.e. with" by "i.e., with". | Done. |
| Page 2, line 19: Replace "by (Park et al., 2014)" by "by Park et al. (2014)". | Done. |
| Page 3, line 1: Replace "13 Aug" by "13 August". | Done. |
| Page 3, line 23: Replace "samples collection" by "sample collection". | Done. |
| Page 4, line 8: Replace "is same" by "is the same". | Done. |
| Page 4, line 16: Replace "for lightest" by "for the lightest". | Done. |
| Page 5, line 22: Replace "TEOM data" by "the TEOM data". | Done. |
| Page 5, line 25: Abbreviations and acronyms (here "DAQ") should be defined (written full-out) when first used. | Done. We replaced "an erroneous DAQ value" with "software malfunction" |
| Page 5, line 29: Replace "e.g. according" by "e.g., according". | Done. |
| Page 5, line 31: Replace "value where" by "value, which were". | Done. |
| Page 5, line 39: Replace "1 Aug" by "1 August". | Done. |
| Page 6, line 27: Replace "e.g. blank" by "e.g., blank". | Done. |
| Page 7, line 11: Replace "to Gerboles" by "to those in Gerboles". | Done. |
| Page 7, line 20: Replace "e.g. an" by "e.g., an". | Done. |
| Page 7, line 24: Replace "i.e. of" by "i.e., of". | Done. |
| Page 7, line 25: Replace "Hg were" by "Hg, which were". | Not agreed. |
| Page 7, line 32: Replace "Jul and 1 Aug" by "July and 1 August". | Done. |
| Page 7, line 37: Replace "of these two elements reflect" by "of Sb and Sn reflect". | Done. |
| Page 8, line 21: Replace "2 Aug" by "2 August". | Done. |
| Page 8, line 24: Replace "e.g. at" by "e.g., at". | Done. |
| Page 8, line 33: Replace "1 Aug" by "1 August". | Done. |
| Page 9, line 2: Insert a space between "period," and "the". | Done. |
| Page 9, line 3: Replace "e.g." by "e.g.,". | Done. |
| Page 9, line 7: Replace "e.g. elemental" by "e.g., elemental". | Done. |
| Page 9, line 19: Replace "2 Aug" by "2 August". | Done. |
| Page 9, line 24: Insert a space before "µg". | Done. |
| Page 9, line 26: Replace "in brackets" by "in parentheses". | Done. |

Page 9, line 37: Abbreviations and acronyms (here "eBC") should be defined (written full-out) when first used.                                                    Done.
Page 10, line 2: Replace "e.g. Sr" by "e.g., Sr".                    Done.
Page 10, line 7: Replace "e.g. Hopke" by "e.g., Hopke".              Done.
Page 10, line 25: Replace "inversion. . It" by "inversion. It".      Done.
Page 11, line 14: Replace "comprised of approximately" by "comprised approximately". Done.
Page 11, line 19: Replace "Measured concentrations" by "The measured concentrations". Done.
Page 11, line 23: Replace "dependent on elements" by "depending on the element". Done.
Page 11, line 38: Replace "e.g. 2" by "e.g., 2".                     Done.
Page 12, line 5: Replace "Continuous" by "The continuous".           Done.
Page 12, line 24: Replace "M.C.Minguillón" by "M.C. Minguillón".     Done.
Pages 12-15, Reference list:
- abbreviated journal names should be used throughout;              Checked.
- titles of journal articles should be in lower case and not in Title Case.    Checked.
Page 15, lines 39-42: Yatkin et al. (2012) should come before Yatkin et al. (2016). Done.
Page 17, Figure 2: The text in the right ordinate does not come out properly.    Figure exchanged.
Page 19, legend of Figure 4: Replace "elemnt" by "element".          Done. Also
removed hyphen in Xact-625.
Page 19, caption of Figure 4: Replace "Xact625" by "Xact 625".       Done.
Page 20, ordinate of Figure 5: Replace "Xact625" by "Xact 625".      Done.
Page 28, footnote 2) of Table 2: Replace "Gälli" by "Gälli Purghart".    Done.
Page 28, footnote 7) of Table 2: "Alastuey et al. 2016" is not in the Reference list. Done.

For the Supplement:
Page 1, line 2: Replace "Table 2)" by "Table 1)".          Done.
Page 1, line 16: Replace "47mm" by "47 mm".                Done.
Page 1, line 22: Replace "Table S1" by "Table S2".         Done.
Page 2, line 22: Replace "Table S2" by "Table S3".         Done.
Page 3, line 4: Replace "Table S2" by "Table S3".          Done.
Page 3, line 7: Replace "Tanner et al. 1974" by "Tanner et al., 1974".     Done.

[revised manuscript text omitted]
 19 %. Similarly the agreement between each of the labs and bench top XRF is good as well. If Se and Ca are excluded the average percent difference between XRF and IDEA is 5.4 % while the difference between XRF and

ERG is -3.1 %. The comparison of the daily averaged Xact values with the benchtop XRF values shows an average difference of 37 % (Xact-CES)/CES) for the elements Zn, Cu, Fe, K, Ca, and Mn, which is close to the observed mean difference to ICP. It is also consistent in the sense that all average differences Xact – CES for these elements are positive. The benchtop XRF and the Xact are typically within 5 % when analysing the same standard. Further both benchtop XRF and Xact use the same type of fitting routine (with minor differences in the determination of spectral background), hence the most likely explanation for the difference between the Xact and the labs is differences due to sampling or sampling location.

**S3. Spiked filter samples for method intercomparisons**

[revised manuscript text omitted]

10 A statistical analysis revealed that at the 99 % confidence level only Cu showed a significant difference between the two filter types. ICP showed differences between -17 % and +32 % (average 5.6 %) for IDAEA-CSIC, and -7 % and +58 % (average 15.6 %) for ERG for quartz filters. The scatter is much larger than for the field samples, and differences can be positive or negative.

**S4. Diurnal variations of elements for fireworks and non-fireworks periods**

[Figure]

**Figure S4: Diurnal variations of the Group A elements Si, S, Cl, K, Ti, Mn, Fe, Cu, Zn, and Pb. See Fig. 6.**

**S5. Diurnal variations of elements for north and south wind sectors**

[Figure]

**Figure S5: Diurnal variations of the Group A elements Si, Cl, Ca, Ti, Fe, Cu, Zn, and Ba. South means a wind from the freeway towards the station. See Fig. 8.**

5  **Reference**

Tanner, T. M., Young, J. A., and Cooper, J. A.: Multielement analysis of St. Louis aerosols by nondestructive techniques, Chemosphere, 3, 211-220, 1974.

---

## Author Response (AR3)

**Response to the Associate Editor**

**Comments to the Author:**

One alteration is still needed before the manuscript can be published in AMT.

5

For the main text:

Page 13, line 4: Replace "Atmospheric Environment. Part A. General Topics, 26" by "Atmos. Environ. Part A, 26".

**10 Authors' reply:**

We changed the reference as requested.

I would like to thank the Associate Editor for his careful effort and patient support.

**Elemental composition of ambient aerosols measured with high temporal resolution using an online XRF spectrometer**

Markus Furger1,\*, María Cruz Minguillón2, Varun Yadav3, Jay G. Slowik1, Christoph Hüglin4, Roman Fröhlich1, Krag Petterson3, Urs Baltensperger1, André S. H. Prévôt1

[revised manuscript text omitted]